# Chemoenzymatic synthesis of 3-ethyl-2,5-dimethylpyrazine by L-threonine 3-dehydrogenase and 2-amino-3-ketobutyrate CoA ligase/L-threonine aldolase

Tomoharu Motoyama [1], Shogo Nakano [1,2✉], Fumihito Hasebe [1], Ryo Miyata[1], Shigenori Kumazawa[1], Noriyuki Miyoshi[1] & Sohei Ito [1✉]

Pyrazines are typically formed from amino acids and sugars in chemical reactions such as the Maillard reaction. In this study, we demonstrate that 3-ethyl-2,5-dimethylpyrazine can be produced from L-Thr by a simple bacterial operon. We conclude that EDMP is synthesized chemoenzymatically from L-Thr via the condensation reaction of two molecules of aminoacetone and one molecule of acetaldehyde. Aminoacetone is supplied by L-threonine 3-dehydrogenase using L-Thr as a substrate via 2-amino-3-ketobutyrate. Acetaldehyde is supplied by 2-amino-3-ketobutyrate CoA ligase bearing threonine aldolase activity from L-Thr when CoA was at low concentrations. Considering the rate of EDMP production, the reaction intermediate is stable for a certain time, and moderate reaction temperature is important for the synthesis of EDMP. When the precursor was supplied from L-Thr by these enzymes, the yield of EDMP was increased up to 20.2%. Furthermore, we demonstrate that this reaction is useful for synthesizing various alkylpyrazines.

[1] Graduate School of Integrated Pharmaceutical and Nutritional Sciences, University of Shizuoka, Shizuoka, Japan. [2] PREST, Japan Science and Technology Agency, Tokyo, Japan. ✉email: snakano@u-shizuoka-ken.ac.jp; itosohei@u-shizuoka-ken.ac.jp

Pyrazines are nitrogen-containing heterocyclic aromatic compounds broadly found in nature, especially in foods such as coffee, chocolate, soybeans, and thermally processed foods. Pyrazine and its derivatives are not only responsible for the flavor of roasted foods, but also act as chemical transmitters of living organisms. For example, pyrazine analogs (2,6-dimethylpyrazine, 3-ethyl-2,5-dimethylpyrazine [EDMP], and 2,3,5-trimethylpyrazine) are highly present in wolf urine and induce fear-related responses in deer[1,2], rats[3], and mice[4]. As well, they act as trail and alarm pheromones in leaf-cutter ants[5]. From the perspective of medical and clinical applications, some pyrazine compounds have the potential to treat some diseases. For example, aloisine A is utilized as a therapy for Alzheimer's disease[6], pyrazinamide is used as an anti-tuberculosis drug[7], and favipiravir is used as an anti-RNA antiviral drug targeting influenza and coronavirus[8].

Based on these considerations, it is no doubt that pyrazines are important compounds for both basic and applied studies. This raises the question of how pyrazines are synthesized under both chemical and physiological conditions. Many researchers have attempted to elucidate the synthetic pathway of pyrazines. Stepwise reactions catalyzed by the Maillard reaction and Strecker degradation are the well-known processes used to synthesize pyrazines chemically[9,10]. In the Maillard reaction, an aldehyde group of a reducing sugar and an amino group of an amino acid form a Schiff base by heating a reducing sugar and an amino acid, and then an α-dicarbonyl compound is generated through Amadori transfer and a ketoamine. Subsequently, the α-dicarbonyl compound reacts with the free amino acids, and $CO_2$, aldehydes, and ethanolamines are generated by Strecker degradation. Finally, the condensation of ethanolamine produces pyrazines. On the other hand, the biosynthetic pathway of pyrazines is somewhat complicated; many enzymes contribute to producing the pyrazines. For example, Masuo et al. reported that 4-aminophenylalanine (4APhe) is produced by the papABC enzymes, and 2,5-dimethyl-3,6-bis(4-aminobenzyl)pyrazine is produced from two molecules of 4APhe by papDEF enzymes in *Pseudomonas fluorescens* SBW25[11]. As for another pathway, only qualitative analysis has been performed to this point. For example, *Bacillus subtilis* IFO 3013 produces 2,5-dimethylpyrazine (DMP) when cultured under L-threonine (L-Thr)-rich conditions[12,13]. However, until recently, no enzyme involved in pyrazine biosynthesis from L-Thr has been reported, and its synthesis mechanism had remained unknown.

Recently, the biosynthetic pathway to produce pyrazines from L-Thr was suggested by several research groups. Papenfort et al. reported that 3,5-dimethylpyrazine-2-ol (DPO), which controls biofilm formation, was produced by metabolizing L-Thr in *Vibrio cholerae*. This group reported that aminoacetone, which is produced by L-threonine 3-dehydrogenase (TDH) could be used as a component in the synthesis of DPO[14]; DPO was synthesized through the condensation and cyclization of aminoacetone and *N*-alanyl-aminoacetone using many unassigned enzymes[15]. In addition, aminoacetone and metabolites of D-glucose are precursors to synthesize alkylpyrazines in *Bacillus subtilis* 168[16,17]. These reports suggest that by utilizing aminoacetone as a basic structure to synthesize alkylpyrazines, there are many biological reaction pathways which synthesize pyrazines, and many new enzymes and methods which could be explored in pyrazine synthesis.

In this study, we attempted to discover a synthetic pathway in bacteria that could produce alkylpyrazine from only L-Thr. To find this pathway, we focused on an unexplored but conserved bacterial operon formed by the *tdh* and *kbl* genes, which encode TDH and 2-amino-3-ketobutyrate CoA ligase (KBL), respectively. TDH is an $NAD^+$-dependent enzyme that catalyzes the dehydrogenation of the side chain hydroxy group of L-Thr, converting L-Thr to 2-amino-3-ketobutyrate (AKB)[18–20]. KBL catalyzes the conversion of AKB to glycine (Gly) using CoA[21]. In this study, we show that EDMP can be synthesized from only L-Thr by combinational usage of TDH and KBL. Using X-ray crystallography and biochemical analysis of KBL from *Cupriavidus necator* (CnKBL, PMID WP_011616818.1), we demonstrated that CnKBL exhibits two activities: (1) KBL activity and (2) threonine aldolase activity. We hypothesize that some alkylpyrazines can be produced under normal conditions (e.g., normal temperature and water systems) by combinations of enzymatic and chemical reactions.

## Results and discussion

**Characterization of TDH and KBL-like genes**. Some microorganisms can use L-Thr as their sole source of carbon and energy. First, the catabolic enzymes are TDH and KBL; L-Thr is converted to Gly and acetyl-CoA by these enzymes. However, there are currently a few reports indicating that reactive AKB is converted to some secondary metabolites, such as pyrazines. By analyzing unexplored but conserved TDH and the adjacent KBL-like gene coded in some bacterial operons, we may obtain information which aids in discovering a new pathway to metabolize AKB. In other research, operon analysis is broadly applied to assign new biosynthetic pathways metabolizing target compounds[22].

The gene cluster around the *tdh* gene in *C. necator*, which encodes CnTDH (PMID WP_010813492.1), is shown in Fig. 1a. This indicates that a KBL-like gene from *C. necator* (*Cnkbl*) was located upstream of *Cntdh*, and formed an operon. Here, the operons may be physiologically important to the metabolite L-Thr in several bacterial strains because this was conserved in some bacterial phyla such as Proteobacteria (*C. necator* and *Vibrio cholerae*), Synergistetes (*Thermanaerovibrio acidaminovorans* DSM 6589), and Firmicutes (*Bacillus subtilis* subsp.) (Supplementary Fig. 1, and Supplementary Table 2 and Supplementary Table 3). Annotations of KBL-like genes include 2-amino-3-ketobutyrate CoA ligase, glycine C-acetyltransferase, and pyridoxal phosphate-dependent acyltransferase (Supplementary Table 3). In a previous study, Schmidt et al. . determined the crystal structure of KBL from *Escherichia coli* (EcKBL), which shares 61% sequence identity with CnKBL. Thus, CnKBL may be a PLP-dependent enzyme with KBL activity.

Next, utilizing L-Thr as the substrate, in vitro qualitative analysis was performed to confirm what compounds were synthesized by enzymes translated from the *tdh* and *kbl* genes. The reaction was performed by adding both CnTDH and CnKBL to the reaction solution containing L-Thr with or without CoA. As expected, CnKBL had KBL activity in solutions containing CoA; the peak derived from acetyl-CoA and Gly was confirmed after the reaction (Fig. 1b, c, red lines). As the concentration of CoA decreased, an unknown compound at 137.1 *m/z* increased (Fig. 1d, red, blue and green lines). Curiously, the production of Gly was still confirmed without CoA (Fig. 1c, green line). Of note, these reactants had a strong nutty aroma. Judging from the *m/z* of LC-MS (Fig. 1d), mass spectrum of GC-MS (Supplementary Fig. 2), and NMR analysis (Supplementary Fig. 3), this compound was identified as EDMP. HPLC analysis showed that the yield of EDMP from L-Thr was ~4%. The yield of EDMP was estimated from Supplementary Fig. 4. The production of Gly without CnTDH and CoA was also confirmed, suggesting that CnKBL directly catalyzes the cleavage of the carbon-carbon bond of L-Thr. (Supplementary Fig. 5). A previous study reported that EDMP could be produced from the metabolite of L-Thr and D-Glucose[16,17]. However, we were unable to find any reports that

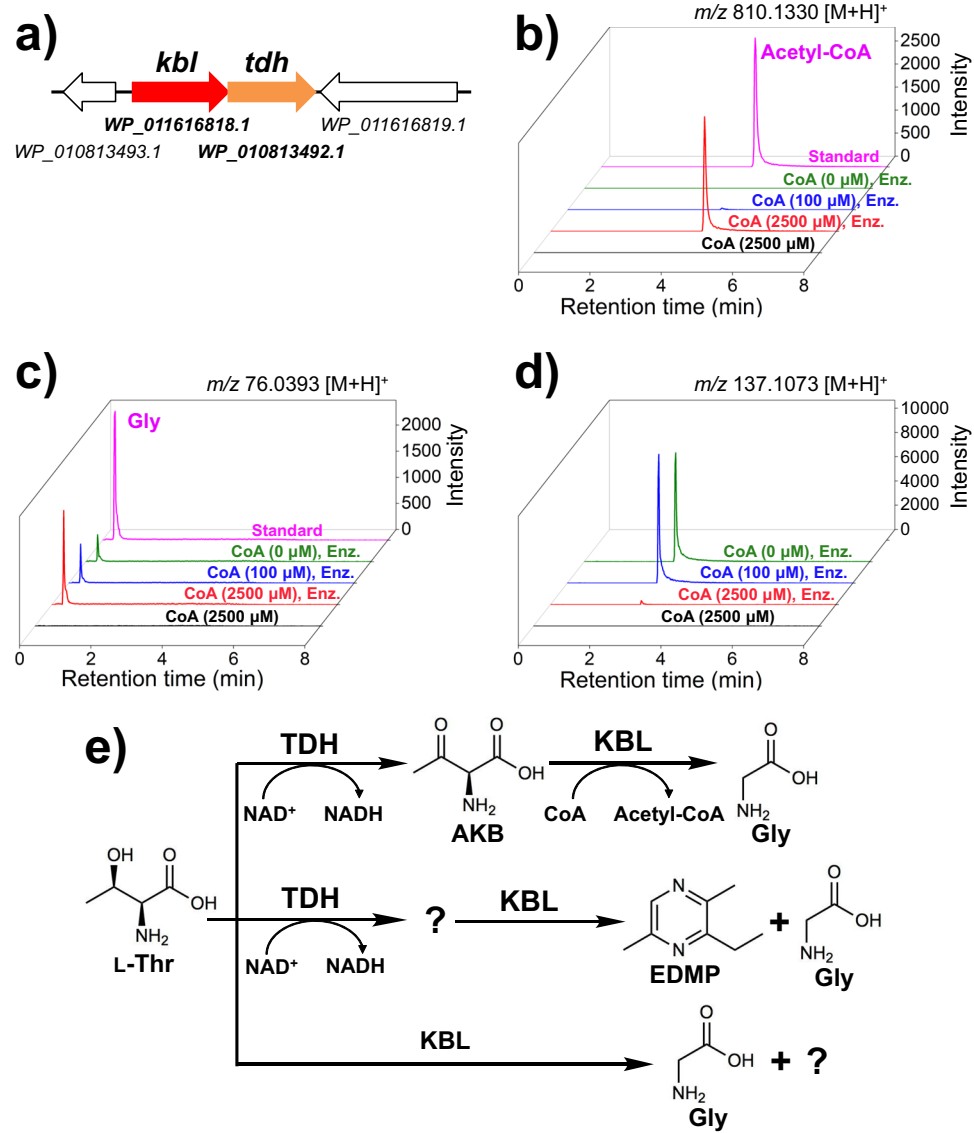

**Fig. 1 Characterization of the *tdh* and *kbl* operon. a** Operon formed by the *tdh* and *kbl* genes in *Cupriavidus necator*. Genes coding *tdh* and *kbl* are colored by orange and red, respectively. LC-HRMS analysis of acetyl-CoA (**b**), Gly (**c**), and EDMP (**d**). The following compounds and enzymes are contained in the reaction system: L-Thr, NAD+, and CoA (2500 μM) (black line); L-Thr, NAD+, CoA (2500 μM), CnTDH, and CnKBL (red line); L-Thr, NAD+, CoA (100 μM), CnTDH, and CnKBL (blue line); L-Thr, NAD+, CnTDH, and CnKBL (green line); and standard (magenta line). Enz. means that CnTDH and CnKBL are contained in the reaction condition. **b** Extracted ion count chromatogram for each compound is as follows: *m/z* of acetyl-CoA, Gly, and EDMP were 810.1330 ± 0.0030 [M + H]+ (**b**), 76.0393 ± 0.0020 [M + H]+ (**c**), and 137.1073 ± 0.0020 [M + H]+, respectively. **e** Schematic reaction pathway to produce EDMP by CnTDH and CnKBL. Utilizing L-Thr as a starting material, CnTDH and CnKBL can generate EDMP.

EDMP can be produced by metabolizing only L-Thr; thus, we believe that this is the first report suggesting that EDMP can be synthesized from only L-Thr.

Next, we used [U-13C,15N]-L-Thr to check whether the supplied L-Thr is incorporated into EDMP by CnTDH and CnKBL (Fig. 2). The increased units of mass (e.g., 10 unit, m/z 145.1–135.1) indicated that all carbon and nitrogen atoms of EDMP were derived from isotope-labeled L-Thr.

How CnTDH and CnKBL metabolize L-Thr is summarized and illustrated in Fig. 1e. We predicted that CnKBL catalyzes not only the acetyltransferase reaction (EC 2.3.1.29), which is broadly recognized by previous studies, but also another reaction. Based on other PLP-dependent enzymes, CnKBL may catalyze the carbon-carbon bond cleavage reaction (EC 4.1.X.X) as a side reaction. This prediction was based on the understanding that TDH would only catalyze the dehydrogenation reaction of L-Thr

or their analogs. This point has been suggested by several previous structural and functional analyses of many TDHs[19,20,23,24]. The production of EDMP was also confirmed by recombinant TDH and KBL-like enzymes (TaTDH and TaKBL) derived from *Thermanaerovibrio acidaminovorans* DSM 6589 (Supplementary Fig. 6), suggesting that the pathway for EDMP production is conserved in some bacterial phyla. Besides utilizing L-Thr as a substrate, EDMP could be synthesized under physiological conditions in *C. necator*. GC-MS analysis indicated that there is a peak corresponding with EDMP after 3 days of cultivation in M9 medium containing L-Thr but not in NBRC medium (Fig. 3). This suggests that EDMP would be synthesized under oligotrophic conditions in this strain.

**Characterization of carbon-carbon bond cleavage reaction of CnKBL.** Other researchers have reported that EDMP can be

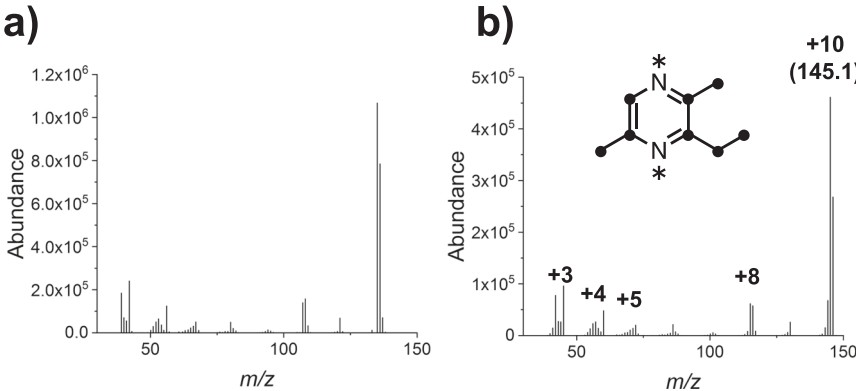

**Fig. 2 EDMP production using [U-¹³C,¹⁵N]-L-Thr.** GC-MS spectra of synthesized EDMP using unlabeled L-Thr (**a**) and [U-¹³C,¹⁵N]-L-Thr (**b**). ¹³C and ¹⁵N atoms are shown as circles and asterisks, respectively.

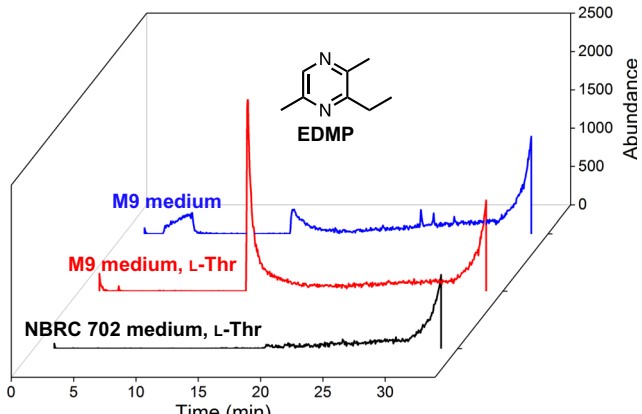

**Fig. 3 EDMP production by *C. necator*.** GC-MS chromatogram of EDMP produced by *C. necator* in NBRC 702 medium supplemented with L-Thr (black) or M9 minimal medium with or without L-Thr (red and blue, respectively).

synthesized chemically from two compounds, aminoacetone and acetaldehyde[25,26]. Based on this, there is a possibility that these compounds could be produced in the reaction system containing CnTDH and CnKBL, and EDMP synthesis would be an end product. The supply of aminoacetone could be easily predicted from previous studies; aminoacetone is produced by the chemical degradation of unstable AKB which is a product of TDH catalysis[21,27].

On the other hand, for the supply of acetaldehyde, there is no conceivable pathway that involves a reaction mechanism of these two enzymes. Because the reaction system contains only L-Thr as a substrate, there is no doubt that acetaldehyde is provided by degrading L-Thr with these enzymes. Among the carbon-carbon bond cleavage enzymes, we focused on aldehyde-forming lyase (EC 4.1.2). L-threonine aldolase (TA, EC 4.1.2.5) and low-specificity L-threonine aldolase (EC 4.1.2.48) are capable of metabolizing L-Thr with the formation of Gly and acetaldehyde. Aldehyde-forming lyases (EC 4.1.2) are PLP-dependent enzymes, as is KBL, and therefore we hypothesized that CnKBL has TA activity.

The TA activity of CnKBL was evaluated by a two-enzyme reaction system containing CnKBL, alcohol dehydrogenase (ADH), and the cofactor NADH (Supplementary Fig. 7). The system can detect acetaldehyde produced by degrading L-Thr with CnKBL; several aldehydes can be reduced by ADH, and TA activity can be estimated by measuring the decrease in NADH by UV-Vis spectrometer. First, to judge whether CnKBL has TA

activity, qualitative analysis was performed by measuring the UV-Vis spectra change in the region ranging from 280 to 600 nm. For this analysis, five amino acids (L-Thr, L-allo-Thr, DL-3-Hydroxynorvaline [DL-3-HN], D-Thr, and L-Ser) were utilized as substrates (Fig. 4a and Supplementary Fig. 8). In three of five substrates (L-Thr, L-allo-Thr, and DL-3-HN), consumption of NADH could be confirmed. Here, the production of Gly, another product synthesized by the TA reaction, was already confirmed by LC-MS analysis (Supplementary Fig. 5), suggesting that CnKBL could catalyze an aldehyde-forming carbon-carbon bond cleavage reaction.

Next, enzyme kinetics analysis of CnKBL for TA activity was performed using L-Thr, L-allo-Thr, and DL-3-HN as substrates. For all of these substrates, the initial velocity for the TA activity of CnKBL could be plotted on the Michaelis-Menten model (Fig. 4b), suggesting that the activity is an enzymatic reaction. The kinetics parameters estimated by fitting the initial velocity to the model are listed in Table 1. The turnover rate for the TA activity of CnKBL was comparable to that of *Streptomyces coelicolor* TA[28], but was much lower than that of *E. coli* TA (EcTA)[29]. The catalytic efficiency ($k_{cat}/K_m$) of CnKBL toward L-Thr was 40-fold lower than that of EcTA. Next, we verified KBL activity. Due to the instability of the AKB substrate and the difficulty in detecting products, Mukherjee et al. used glycine as a substrate and detected the formation of the thiol group using Ellman's reagent (Supplementary Fig. 9)[29]. Therefore, the kinetic parameters of condensation activity in CnKBL were determined and compared with those of EcKBL[29]. The $k_{cat}$, $K_m$, and catalytic efficiency values of CnKBL for Gly were quite comparable to those of EcKBL. The apparent KBL activity of CnKBL is much higher than the TA activity of CnKBL. However, the $K_m$ value of CnKBL was 4.1 times higher than that of EcKBL, suggesting that its activity is susceptible to the concentration of CoA (Supplementary Fig. 10 and Supplementary Table 4). In fact, in low nutrient conditions, the concentrations of CoA and acetyl-CoA in animal and bacterial cells have been reported to be reduced to 1.3–20 µmol[21,30]. These results are consistent with EDMP production in vivo and the discussions mentioned above.

Taken together, we demonstrated that the KBL-like enzymes from *C. necator* and *T. acidaminovorans* catalyze the following two reactions: (1) KBL activity (EC 2.3.1.29) as the primary reaction, and (2) threonine aldolase activity (EC 4.1.2.48) as a side reaction.

**Overall and active-site structure of CnKBL.** Enzyme functional analysis of CnKBL indicated that this enzyme changed gradually from acetyltransferase to aldolase depending on CoA concentration (Fig. 1). Here, the reaction mechanism of KBL activity

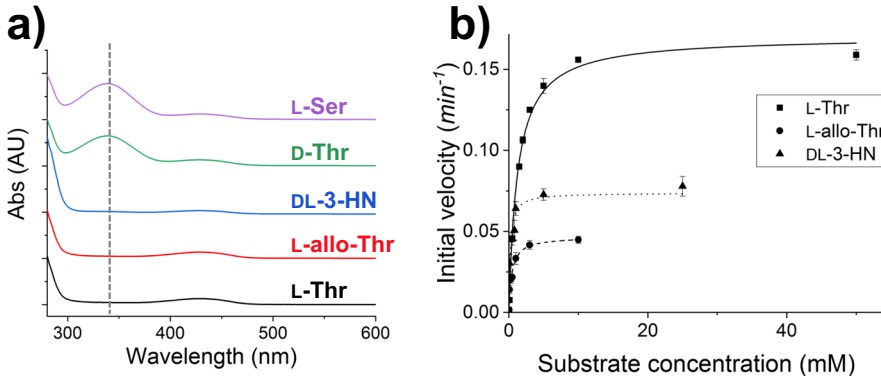

**Fig. 4 Characterization for enzymatic functions of CnKBL. a** UV-Vis spectra analysis measuring NADH consumption under reaction conditions containing CnKBL and substrates. The substrates used for the analysis are as follows: L-Thr (black line), L-allo-Thr (red line), DL-3-HN (blue line), D-Thr (green line), and L-Ser (purple line). **b** Enzyme kinetic plots of TA activity of CnKBL. The initial velocity toward L-Thr, L-allo-Thr, and DL-3-HN are shown as squares, circles, and upward-pointing triangles, respectively. The data are represented as mean ± standard deviation. Kinetic parameters are listed in Table 1.

| Table 1 Enzymatic properties of CnKBL. | | | |
|---|---|---|---|
| Substrate | WT | | |
| | $k_{cat}$ (min$^{-1}$)[a] | $K_m$ (mM)[b] | $k_{cat}/K_m$ (min$^{-1}$ mM$^{-1}$) |
| L-Thr | 0.170 ± 0.39 | 1.24 ± 0.11 | 0.14 |
| L-allo-Thr | 0.0467 ± 0.0026 | 0.42 ± 0.086 | 0.11 |
| DL-3-Hydroxynorvaline | 0.0738 ± 0.0043 | 0.20 ± 0.056 | 0.37 |

[a]$k_{cat}$ values of CnKBL were derived from the data shown in Fig. 4b.
[b]$K_m$ values represent the Michaelis constant value toward L-Thr.

for CnKBL is identical to that of the previously reported EcKBL[21,30]. As supporting data, the sequence identity between EcKBL and CnKBL is high (61%) and active site residues S187 (S185 in EcKBL) and H215 (H213 in EcKBL) are conserved (Supplementary Fig. 11). Furthermore, CnKBL and EcKBL both have a cavity to bind CoA to the active site[21]. However, for the TA activity of CnKBL, no reaction mechanism has previously been proposed. X-ray structural analysis of CnKBL would be helpful to elucidate the molecular mechanism of TA activity. In this study, we first determined the crystal structure of the ligand-free form of CnKBL at a 1.8 Å resolution. CnKBL forms a dimer as well, as already reported with KBL (PDB entry 1FC4) (Fig. 5a). The overall structure of CnKBL is almost identical to 1FC4, and in fact, the root mean square deviation value for $C_\alpha$ atoms between CnKBL and 1FC4 was 0.596 Å (Fig. 5b). The active site structure of CnKBL(WT) is indicated in Fig. 5c. As shown in the electron density map, only PLP is coordinated at the site by forming a Schiff base with the side chain of K246 (Fig. 5c). In addition, there is a cavity at the active site which is formed by seven residues: N52, V81, H138, H215, L274, F275, and R370. These residues contribute to coordinating substrates at the active site and to catalyzing the TA activity of CnKBL. Substrates bind to the cavity and this activates the TA activity.

Here, substrate- and product-bound structures of CnKBL are required to elucidate the reaction mechanism of the TA activity of CnKBL, however, these structures cannot be obtained by utilizing wild-type CnKBL. To clear this hurdle, we attempted to prepare a K246A variant of CnKBL. This variant was designed to prevent the Schiff base exchange reaction and stabilize to a linkage between the PLP cofactor and substrate/product (Supplementary Table 4). As expected, we succeeded in determining the three X-ray crystal structures for the substrate- or product-binding form of CnKBL(K246A) at 2.5 Å (L-Thr), 2.55 Å (L-3-HN), and 2.5 Å

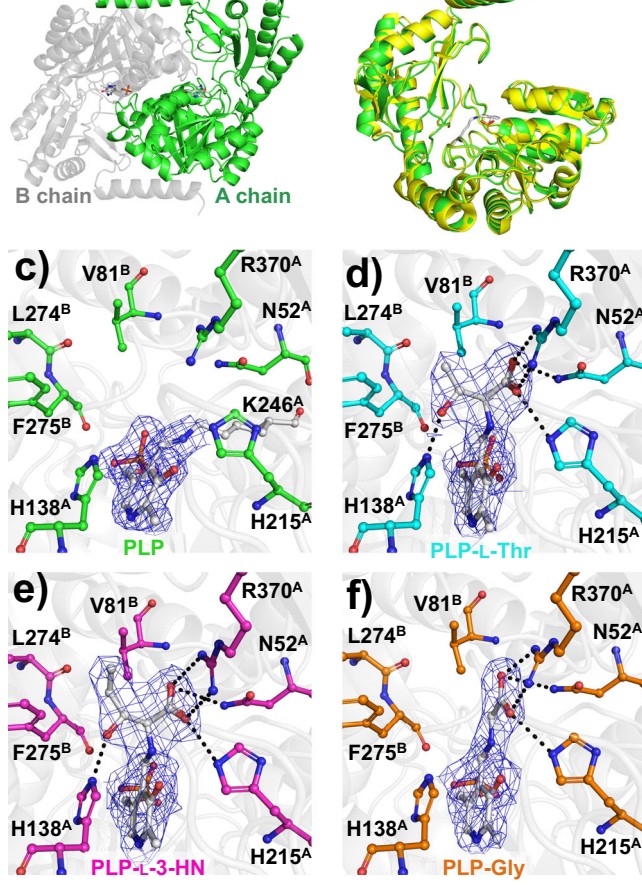

**Fig. 5 Crystal structures of CnKBL(WT) and K246A variants. a** Overall structure of CnKBL. The dimer is shown as green and gray. PLP is shown as a stick model. **b** Superposed structures of CnKBL (green) and EcKBL (yellow, PDB ID: 1FC4). Active site structures of CnKBL(WT) and the K264A variant. Here, structures of the ligand-free form of CnKBL(WT), L-Thr, L-3-HN, and Gly binding forms of CnKBL(K264A) variants were represented as green (**c**), cyan (**d**), magenta (**e**), and orange (**f**), respectively. The hydrogen bonds formed between CnKBL and substrates were represented as dotted lines when the bond length was less than 3.4 Å. The 2Fo-Fc electron density map (blue mesh) is contoured at 1.0σ.

**Fig. 6 Proposed TA reaction mechanism of CnKBL. I** Indicates internal aldimine, (**II**) indicates external aldimine with L-Thr, (**III**) indicates quinonoid intermediate, and (**IV**) indicates external aldimine with Gly, respectively.

(Gly), respectively. The active site structures of the L-Thr (Fig. 5d), L-3-HN (Fig. 5e), and Gly (Fig. 5f) binding forms indicated that the carboxyl group of ligands form a hydrogen bond with the side chains of N52, H215, and R370. The residues are important to fix the ligands at the active site correctly. In addition, the side chain of H138 formed a hydrogen bond with the hydroxyl group of the substrates (Fig. 5d, e) suggesting that H138 could be a potential catalytic residue for TA activity of CnKBL. Indeed, TA activity of the H138F variant decreased to an undetectable level compared with CnKBL(WT) (Supplementary Table 5).

Next, we considered why TA activity of CnKBL was weak compared to eTA. Salvo et al. reported that R169 and R308 contribute to the stabilization of the transition state during the enzymatic reaction in eTA[29,31]. Although R370 in CnKBL, corresponding to R308 in eTA, is conserved, R169 in eTA is mutated to H215 in CnKBL (Supplementary Fig. 12). From this, we considered that the weak TA activity of CnKBL occurs because the surrounding environment, which includes H215, is not optimized to exhibit high TA activity. In addition, Salvo et al. reported that H126 and K222 are important for the arrangement of the water molecule at the active site, which is important for the TA reaction. On the other hand, these residues are mutated to different amino acids in CnKBL (Supplementary Fig. 12). Therefore, we considered that the mutations of these residues in CnKBL also contribute to the weak activity of CnKBL

Summarizing all of the results, we proposed the putative reaction mechanism of TA activity for CnKBL as shown in Fig. 6. The reaction mechanism of TA activity for CnKBL is similar to that for conventional TA[29,32]. First, PLP forms a Schiff base with K246 of CnKBL (internal aldimine) (Fig. 6, I). When the substrate L-Thr comes into the active site, PLP forms a Schiff base with the amino group of L-Thr (external aldimine) (Fig. 6, II). The

catalytic residue H138 catalyzes the dehydrogenation of the hydroxyl group on the side chain of L-Thr (Fig. 6, II). After completion of the retro-aldol reaction, the product acetaldehyde would be released from the active site and PLP forms a Schiff base with the product Gly (quinonoid intermediate) (Fig. 6, III). Thereafter, when the α-carbon of Gly is protonated by a base such as water, an external aldimine is formed between PLP and Gly (Fig. 6, IV). Finally, Gly is released from the active site, and the internal aldimine is formed again (Fig. 6, I).

**Comparison of EDMP production under chemical and chemoenzymatic reactions.** Structural and functional analysis of CnTDH and CnKBL suggested that EDMP could be synthesized only from L-Thr with chemoenzymatic reaction. Aminoacetone and acetaldehyde, precursors to generate EDMP, are supplied by enzymatic reactions, and chemical reactions progress their condensation. The next challenge is to optimize the reaction conditions to maximize the amount of EDMP produced. It is expected that reaction temperature and timing of addition of precursor would affect the rate of EDMP production.

First, we estimated the optimal temperature for the maximum synthesis of EDMP by changing the reaction temperature (Entry 1–4 in Supplementary Table 6). The yield of EDMP was as high as 16.2% under the chemical reaction condition of 30 °C. Notably, this yield gradually decreased as the temperature increased. In the Maillard reaction, pyrazine is generated at high temperatures[9,10]. The fact that the chemical condensation reaction of aminoacetone and acetaldehyde proceeds under such moderate conditions may have implications for the identification of pyrazine compounds in vivo (Fig. 3).

The synthesis mechanism of EDMP is shown in Fig. 7a. First, as is well known, aminoacetone is supplied from L-Thr by TDH. Next, the two aminoacetones condense to 2,5-dimethyl-3,6-dihydropyrazine (DHP). Finally, DHP tautomerizes and undergoes nucleophilic addition to acetaldehyde to synthesize EDMP. Other research groups reported that the DHP is easily oxidized under mild condition to form 2,5-dimethylpyrazine (DMP)[33,34], suggesting that immediate reaction of DHP with acetaldehyde increases the production of EDMP. To confirm this point, we compared production rates of EDMP and DMP by changing the timing of acetaldehyde addition (Supplementary Table 6). As expected, yield of EDMP is maximized at the condition which mixed acetaldehyde and aminoacetone simultaneously (16.2%, entry 5 in Supplementary Table 6). By delaying the timing of addition, the production rate gradually decreased. When acetaldehyde was added after 12 h of pre-reaction, the EDMP production rate was minimal (Entry 9 in Supplementary Table 6). On the other hand, the opposite tendency was observed for the production rate of DMP (Entry 5–9 in Supplementary Table 6 and Supplementary Figs. 13 and 14).

Considering the time-dependent changes in the production rates of EDMP and DMP, it was found that the reaction intermediate, DHP, was stable for a certain time and that the reaction of DHP and acetaldehyde at physiological temperature was important for the chemoenzymatic synthesis of EDMP. When the precursor was supplied from L-Thr by the enzymes, the yield of EDMP was increased up to 20.2%.

**GC-MS analysis of the chemoenzymatically synthesized pyrazines.** Enzyme functional and structural analysis of CnKBL suggested that L-Thr can be degraded into both aminoacetone and acetaldehyde under the reaction condition containing CnTDH and CnKBL, and these two compounds could be precursors to synthesize EDMP. Here, we predicted that the condensation of two aminoacetones and acetaldehyde would progress by following the

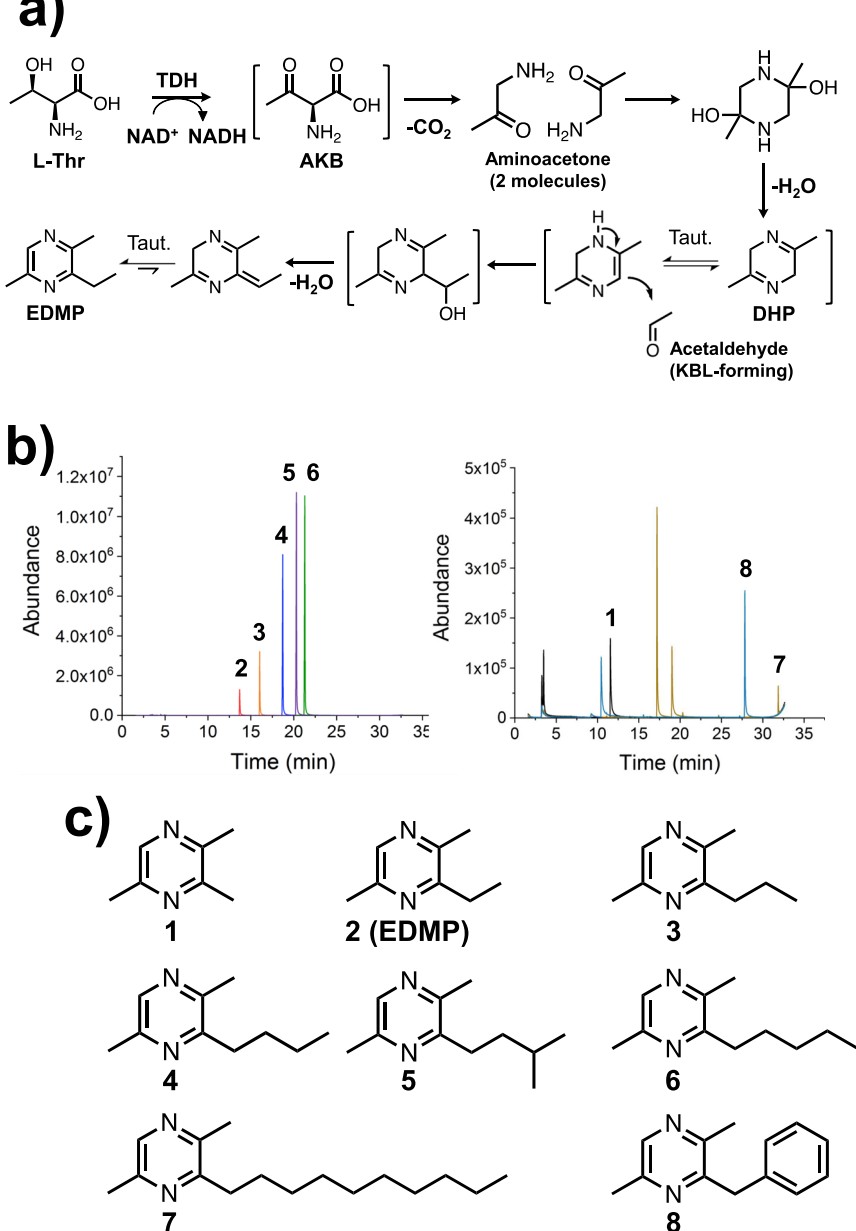

**Fig. 7 Chemoenzymatic synthesis of alkylpyrazines under conditions containing aminoacetone, which is provided by CnTDH and aldehydes. a** Reaction scheme of the chemoenzymatic production of EDMP. Taut. indicates tautomerization. **b** GC-MS chromatograms of synthesized alkylpyrazines. The peaks can be detected in the reaction conditions containing the following aldehydes: formaldehyde (black), acetaldehyde (red), propionaldehyde (orange), butyraldehyde (blue), isovaleraldehyde (purple), valeraldehyde (green), decylaldehyde (yellow), and benzaldehyde (cyan). **c** A list of pyrazines produced from chemoenzymatic reactions is shown as 1–8.

reaction scheme as shown in Fig. 7a. Aminoacetone supplied by CnTDH would be condensed and dehydrated to DHP; the production of DHP has already been reported by other groups[5,16]. EDMP would be produced through tautomerization, the additional reaction of DHP and acetaldehyde, and the subsequent dehydration reaction. If the proposed pyrazine synthesis is plausible, we could easily synthesize various pyrazines by changing the types of aldehydes. To confirm this point, we attempted to synthesize various alkylpyrazines by changing the aldehydes. Here, we utilized the following eight aldehydes in the reaction: formaldehyde, acetaldehyde, propionaldehyde, butyraldehyde, isovaleraldehyde, valeraldehyde, decylaldehyde, and benzaldehyde. Reacted samples were analyzed by GC-MS and the MS spectra for each respective product are indicated in Figs. 7b, c, and Supplementary Fig. 15–22.

In all cases, alkylpyrazines where each aldehyde is incorporated into the products, were produced as expected (Fig. 7b compounds 1–8). The synthesized pyrazines, which are identified by the GC-MS spectra library, are listed in Fig. 7c. For the synthesis of the EDMP, the condensation reaction can be progressed under the existing conditions of a low concentration of aminoacetone and acetaldehyde. In fact, EDMP can be produced under conditions containing 500 nM and 5 μM of acetaldehydes (Supplementary Fig. 23). In human blood, ~1.3 μM of acetaldehydes is consistently circulating[35]. Therefore, it is not impossible to produce alkylpyrazines under physiological conditions when there are hundreds to thousands nM of aldehydes and aminoacetone.

Summarizing these results, various alkylpyrazines can be synthesized by following the chemoenzymatic reaction shown

**Fig. 8 Proposed synthesis pathway of EDMP from L-Thr using TDH and KBL enzymes.** TDH converts L-Thr to AKB and AKB is metabolized to Gly by the acetyltransferase activity of KBL in usual conditions. However, in poor nutrition and at low CoA concentrations, unstable AKB is gradually decarboxylated to aminoacetone (red) and the lyase activity of KBL decomposes L-Thr to acetaldehyde and Gly (blue). EDMP is synthesized chemoenzymatically from L-Thr via the condensation reaction of two molecules of aminoacetone and one molecule of acetaldehyde.

in Fig. 7a, even under the conditions of low concentrations of aldehydes and aminoacetone. This suggests that several of alkylpyrazines could be produced under physiological conditions, such as products 2, 5, and 6 which act as pheromones in the leaf-cutter ant[5].

## Conclusion

In this study, we demonstrate that primary and secondary metabolites, namely Gly and EDMP, can be produced from only L-Thr by simple bacterial operons containing *tdh* and *kbl*. Biochemical and structural analysis of the enzymes indicated that CnKBL exhibits acetyltransferase (EC 2.3.1.29) and aldehyde-forming lyase (EC 4.1.2.48) activities. TDH converts L-Thr to AKB and AKB is metabolized to Gly by the acetyltransferase activity of KBL in usual conditions (Fig. 8). However, in poor nutrition and at low CoA concentrations, unstable AKB is gradually decarboxylated to aminoacetone and the lyase activity of KBL decomposes L-Thr to acetaldehyde and Gly (Figs. 7a and 8). The next challenge will be to optimize this synthetic method to achieve maximum yields (for example,>50%) of alkylpyrazines utilizing L-Thr as a substrate. We are now trying to develop a method to accomplish this by combining chemoenzymatic reactions.

## Methods

**Site-directed mutagenesis of CnKBL variants**. Dried cells of *C. necator* (NBRC 102504) were purchased from NBRC (Biological Resource Center, NITE). The gene for *cnkbl* on the genome of *C. necator* was amplified by PCR using primers (Supplementary Table 1). The amplified gene was subcloned into a pET-28a vector cleaved by *NcoI*/*XhoI*. The prepared pET-28a-*Cnkbl* plasmids were utilized in the following experiment. Site-directed mutagenesis was performed using the Quik-Change lightning Multi-site mutagenesis kit (Agilent Technologies, Santa Clara, CA, USA). Primers used to create the variants are listed in Supplementary Table 1. CnKBL variants were confirmed by DNA sequencing.

**Overexpression and purification of CnKBL(WT) and its variants, TaTDH and TaKBL**. CnKBL and variant plasmids were transformed into the *Escherichia coli* strain BL21(DE3). pET-28a-*Cnkbl* in BL21(DE3) was cultivated overnight at 37 °C in 5 mL of Luria-Bertani (LB) medium containing 30 μg/mL kanamycin. After cultivation on a 5 mL scale, cultured *E. coli* was inoculated into 1 L of LB medium containing 30 μg/mL kanamycin for 4 h at 37 °C. The induction of CnKBL was initiated by adding isopropyl β-D-1-thiogalactopyranoside to a final concentration of 0.5 mM. The culture was cultivated for 18 h at 22 °C. Subsequently, the recombinant cells were collected by centrifugation at 5000 × *g* for 10 min at 4 °C. The overexpressed cells were suspended in buffer A (10 mM potassium phosphate [pH 8.0] and 10 mM NaCl) and sonicated. The insoluble fraction was removed by centrifugation at 11,000 × *g* for 30 min. The supernatant was applied to a 5 mL Ni²⁺-Sepharose column. The column was washed with more than five column

volumes of buffer A. The samples were eluted by a stepwise gradient with three column volumes of buffer A containing either 10, 30, 50, 70, 100, 300, or 500 mM imidazole. Eluted samples were concentrated and applied to a gel filtration column (Superdex 200 increase, GE Healthcare) equilibrated with buffer A. Fractions containing the samples were collected and concentrated for assay and crystallization. Purity was confirmed by sodium dodecyl sulfate-polyacrylamide gel electrophoresis. The gel was stained with Coomassie Brilliant Blue R-250 (Wako). The same procedure was used for CnTDH-pET-15b, TaTDH-pET-15b, and TaKBL-pET-15b.

**Enzymatic reaction using CnTDH and CnKBL**. The enzymatic reaction using CnTDH and CnKBL was performed using assay buffer A (100 mM potassium phosphate [pH 8.0], 10 mM L-Thr, and 5 mM NAD⁺) with 0, 100, or 2500 μM CoA. To initiate the reaction, 0.05 mM CnTDH and CnKBL were added to assay buffer A. The reaction was performed for 3 h at 30 °C. After the enzymatic reaction, the reaction mixture was then subjected to LC-HRMS analysis.

**LC-HRMS analysis**. To detect acetyl-CoA, Gly, and EDMP, LC-HRMS analysis was performed using Q Exactive (Thermo Fisher Scientific, MA), equipped with an electrospray ionization module and the columns described below, which were joined to the LC-HRMS system. For the analysis of Acetyl-CoA, we used a UPLC column {XBridge BEH Amide XP column (length, 2.1 × 50 mm²; inner diameter [i.d.], 2.5 μm; Nihon Waters K.K., Tokyo, Japan)} equipped with a guard column (XBridge BEH Amide XP VanGuard cartridge [length, 2.1 × 5 mm²; i.d. 2.5 μm; Nihon Waters K.K., Tokyo, Japan]). The column was maintained at 40 °C. 5 mM ammonium formate in 90% acetonitrile (solvent A) and 5 mM ammonium formate in 50% acetonitrile (solvent B) were used as mobile phases for the gradient elution of products at a flow rate of 0.4 mL/min. Products were eluted as follows: 0% B for 1 min, 0–100% B over 4 min, 100% B for 2 min, and 0% B for 5 min. For the analysis of Gly and EDMP, we used a UPLC column (XBridge BEH C18 XP column [length, 2.1 × 50 mm²; inner diameter (i.d.), 2.5 μm; Nihon Waters K.K., Tokyo, Japan]) equipped with a guard column (XBridge BEH C18 XP VanGuard cartridge [length, 2.1 × 5 mm²; i.d., 2.5 μm; Nihon Waters K.K., Tokyo, Japan]). The column was maintained at 40 °C. 0.1% formic acid in H₂O (solvent A) and 0.1% formic acid in acetonitrile (solvent B) were used as mobile phases for the gradient elution of products at a flow rate of 0.4 mL/min. Products were eluted as follows: 10% B for 1 min, 10–90% B over 4 min, 90% B for 2 min, and 10% B for 5 min.

**HPLC analysis**. The yield of EDMP produced by CnTDH and CnKBL from L-Thr was estimated by reverse-phase high performance liquid chromatography (HPLC) analyses; here, the Shimadzu apparatus (Prominence) equipped with a UV-vis detector (SPD-20AV, Shimadzu) and Unison UK-C18 column (length, 2 × 150 mm²; i.d., 3 μm; Imtakt Corporation, Kyoto, Japan) were adopted as the detection system. The elution of the compounds was monitored by measuring UV spectra changes at 280 nm. Mobile phase condition was 20% (v/v) acetonitrile and the column was maintained at 40 °C. The flow rate was 0.30 mL/min. The yield of EDMP was estimated from peak area value which is fitted to a calibration curve obtained using EDMP standard. EDMP used as a standard was purchased from AmBeed, Inc. (Arlington Hts, IL, USA).

**Stable isotope tracing experiments**. Stable isotope tracing experiments were performed using assay buffer B (100 mM potassium phosphate [pH 8.0], 10 mM

[U-$^{13}$C,$^{15}$N]-L-Thr, and 5 mM NAD$^+$). To initiate the reaction, 0.05 mM CnTDH and CnKBL were added to assay buffer B. The enzymatic reaction was performed for 3 h at 30 °C. After the enzymatic reaction, the reaction mixture was subjected to GC-MS analysis.

**GC-MS analysis**. Volatile compounds were analyzed by a solid-phase micro-extraction (SPME) method with gas chromatography/mass spectrometry (GC-MS) on Agilent 6890 GC-5975 MSD. In total, 500 μL of the reaction solution was sealed in a 10 mL vial. SPME fiber (PDMS, Sigma-Aldrich) was exposed in the headspace of the vial for 30 min at 50 °C. The fiber was desorbed in the injection port and maintained at 250 °C. GC separation was performed on a capillary column (HP-5MS, 30 m × 0.25 i.d., 0.25 μm film thickness; Agilent) using a temperature gradient of 50–200 at 5 °C/min and 200–280 at 30 °C/min under He gas. Products were identified by comparing the $m/z$ of the sample with the library (NIST 11).

**EDMP biosynthesis by C. necator in vitro**. C. necator was cultivated for 2 days at 30 °C in 5 mL of NBRC 702 medium. The cultured cells were collected by centrifugation at 5000 × g for 10 min at 4 °C, and the cells were resuspended in 5 mL of NBRC 702 and M9 minimal mediums with or without 100 mM L-Thr. The resuspended cells were cultivated for 3 days at 30 °C. Volatile compounds were analyzed by head-space solid-phase microextraction (HS-SPME) with GC-MS, as mentioned above.

**Assay of L-threonine aldolase activity**. The TA activities of CnKBL were measured by quantitating the reduction of NADH using the coupling reaction of yeast alcohol dehydrogenase (Wako) at various concentrations of L-Thr at 30 °C. Enzyme samples were kept in ice-cold water until just before measurements of enzyme kinetics. Enzymatic activity toward L-Thr was measured using assay buffer C (50 mM potassium phosphate [pH 8.0], 0.2 mM NADH, 0.1–50 mM L-Thr, and 50 U yeast alcohol dehydrogenase). To start the reaction, 90 μL of assay buffer C was added to a cuvette, and then 10 μL of the enzyme solution was added. The cuvette was placed in an ultraviolet-visible (UV-vis) spectrometer (UV-2450, SHIMADZU), and the time-dependent reduction of NADH at 340 nm was measured for 3 min. The NADH concentration was calculated using the molar extinction coefficient (6300 M$^{-1}$ cm$^{-1}$) of NADH at 340 nm. The initial velocity at different concentrations of L-Thr was determined and plotted using ORIGIN software. The enzyme kinetic parameters $k_{cat}$ and $K_m$ were determined using the Michaelis-Menten equation and by applying the nonlinear least squares method. The same procedure was performed for enzymatic activity toward L-allo-Thr and DL-3-hydroxynorvaline.

**Assay of 2-amino-3-keto-butyrate CoA ligase activity**. The KBL activities of CnKBL were measured by quantitating the production of CoA using the coupling reaction of 5,5'-dithiobis(2-nitrobenzoic acid) (DTNB) at various concentrations of Gly and CoA at 30 °C[36–38]. Enzyme samples were kept in ice-cold water until just before measuring enzyme kinetics. Enzymatic activity toward Gly and acetyl-CoA was measured using assay buffer D (50 mM potassium phosphate [pH 8.0], 1 mM Acetyl-CoA, 1–200 mM Gly, and 0.1 mM DTNB) and assay buffer E (50 mM potassium phosphate [pH 8.0], 100 mM Gly, 0.01–2 mM acetyl-CoA, and 0.1 mM DTNB), respectively. In total, 90 μL of the assay buffer was added to the cuvette and 10 μL of enzyme solution was subsequently added to initialize the reaction. The cuvette was inserted into a UV-vis spectrometer (UV-2450, SHIMADZU) and the time-dependent production of 5-mercapto-2-nitrobenzoic acid at 412 nm was measured for 3 min. The produced 5-mercapto-2-nitrobenzoic acid concentration was calculated using the molar extinction coefficient (15500 M$^{-1}$ cm$^{-1}$) of 5-mercapto-2-nitrobenzoic acid at 412 nm[36–38]. The initial velocities at different concentrations of Gly and acetyl-CoA were determined and plotted using ORIGIN software. The enzyme kinetic parameters $k_{cat}$ and $K_m$ were determined using the Michaelis-Menten equation and by applying the nonlinear least-squares method.

**Crystallization of binary and ternary forms of CnKBL**. Screening for crystallization of CnKBL was performed using Crystal Screen (Hampton Research, Aliso Viejo, CA), Index (Hampton Research, Aliso Viejo, CA), and Additive Screen (Hampton Research, Aliso Viejo, CA) at 22 °C using the sitting-drop vapor diffusion method. The CnKBL sample was concentrated to 23 mg/mL using the Amicon Ultra-15 centrifugal filter (Millipore). A 1.5 μL sample was mixed with 1.5 μL of each reservoir solution from the screening kit. CnKBL (binary form) crystals appeared in Crystal Screen condition 9 (0.2 M ammonium acetate, 0.1 M sodium citrate tribasic dehydrate [pH 5.6], and 30% [w/v] PEG4000) with Additive Screen condition 80 (40% [v/v] Polypropylene glycol P 400). CnKBL (ternary form) crystals with L-Thr appeared in Index condition 75 (0.2 M lithium sulfate monohydrate, 0.1 M BIS-TRIS [pH 6.5], and 25% [w/v] PEG3350) with Additive Screen condition 54 (30% [w/v] D-(+)-Glucose monohydrate). CnKBL (ternary form) crystals with L-3-hydroxynorvaline appeared in Index condition 75 with Additive Screen condition 57 (30% [w/v] D-Sorbitol). CnKBL (ternary form) crystals with Gly appeared in Index condition 75 with Additive Screen condition 51 (30% [w/v] dextran sulfate sodium salt [Mr 5000]). Ternary complex crystals of CnKBL with L-

Thr, L-3-hydroxynorvaline, or Gly were obtained by adding 10 mM of each substrate to CnKBL. All crystals were quickly soaked in a cryo-reservoir before X-ray data collection. The cryo-reservoir used in the crystals of CnKBL was prepared by the addition of 20% (w/v) PEG400 to each crystallization condition.

**X-ray data collection**. The soaked crystals were mounted and flash-cooled under a nitrogen stream (−173 °C). Diffraction data were collected using a PILATUS3 S6M instrument at BL5A of the Photon Factory (Tsukuba, Japan). The collected data were integrated and scaled using XDS[39] and SCALA. The initial phases were determined by the molecular replacement method with MOLREP[40] using the crystal structure of KBL as a template (PDB entry 1FC4). Model building and structure refinement were performed using COOT[41] and either REFMAC[42] or PHENIX[43], respectively. All figures were prepared with PyMOL[44]. Crystallographic parameters are listed in Table 2.

**Chemoenzymatic reaction using CnTDH and aldehydes**. The chemoenzymatic reaction was performed using 0.5 mL of assay buffer F (100 mM potassium phosphate [pH 8.0], 5 mM L-Thr, 5 mM NAD$^+$, 0.05 mM CnTDH, and 2.5 mM of each aldehyde). The reaction was performed for 12 h at 30 °C. The reaction mixture was then subjected to GC-MS analysis.

**Reporting summary**. Further information on research design is available in the Nature Research Reporting Summary linked to this article.

## Data availability
Protein and DNA sequence data for CnTDH and CnKBL are registered in Supplementary Data 1 and 2, respectively. PDB data for CnKBL are available from the PDB database (PDB ID: 7BXP, 7BXQ, 7BXR, and 7BXS). Validation reports for the PDB data are available in the Supplementary Data 3–6.

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

## Acknowledgements

X-ray data were collected at the synchrotron facilities of the Photon Factory (PF) using beamlines BL5A (proposal No. 2018G006). This work was supported by JSPS KAKENHI Grant Numbers 16K18688 and 18K14391 (for S.N.), 17K06931 (for S.I.), 19J23697 (for T.M.), and by JST, PRESTO Grant Number JPMJPR20AB (for S.N.).

## Author contributions

S.N. and S.I. designed the research and managed the projects. T.M. performed X-ray crystallography, enzyme kinetics analysis, HPLC analysis, GC-MS analysis, and NMR analysis. F.H. and N.M. performed LC-MS analysis. R.M. and S.K. performed LC and NMR analysis. All authors analyzed and discussed the data. T.M., S.N., and S.I. wrote the manuscript.

## Competing interests

The authors declare no competing interests.
