## [Peer Review File · Communications Chemistry]

Reviewers' comments:

Reviewer #1 (Remarks to the Author):

This paper demonstrates that the CnTDH/CnKBL couple convert L-threonine to EDMP. Threonine aldolase (TA) activity of CnKBL participates in the conversion reaction while Marcus et al. have already reported this activity of Escherichia coli KBL (BBA, 1993, 1164, 299-304), which minimizes the impact of this paper. The chemoenzymatic condensation of aldehydes and threonine/aminoacetone to alkylpyrazines may be applicable to pharmaceutical and flavor industries.

The proposed mechanism synthesizing EDMP may be novel. However, I don't think that the results are conclusive enough to show the author's claim, and cannot recommend this article for publication at this stage.

- 1) Stoichiometry of the reaction should be address. For example, reaction yields of EDMP and other pyrazines, which could be produced from the reaction intermediate, aminoacetone, are required. The author's model of the entire reaction (Fig. 6) is hardly acceptable in the absence of these data.
- 2) Is the TA activity high enough to consider its primary role as TA rather than KBL under physiological conditions? Kinetic analysis of KBL may help understanding this.
- 3) Physiological role of the CnTDH-CnKBL operon is poorly supported by the data. Production of EDMP by Cupriavidus bacterium demonstrates conclusive evidence.
- 4) I recommend CnTDH-CnKBL reactions containing isotope-labeled threonine to clarify the 3-ethyl residue of EDMP originating from acetaldehyde.

Minor comment:

Line 110, Origin of the plasmid, and the bacterium should be provided.

Line 310, Missing reference.

Line 356, changes

Figure 5a, Please note molar relationship of the chemicals.

Reviewer #2 (Remarks to the Author):

The present study reports a novel chemo-enzymatic reaction utilizing L-threonine-3-dehydrogenase (TDH) and 2-amino-3-ketobutyrate CoA ligase (KBL) as catalysts and L-Thr as a sole substrate can produce 3-ethyl-2,5-dimethylpyrazine (EDMP). The authors propose that the reaction appears to proceed by delivering aminoacetone ($\text{CH}_3\text{COCH}_2\text{-NH}_2$) and (acetaldehyde + glycine) in the absence of CoA. The authors also reported the formation of EDMP derivatives that differ in the side chain group, which derives from aldehyde substrates added to the chemoenzymatic reactions. The authors also reported crystallographic evidences of enzymatic mechanisms for recognizing PLP-bound L-Thr, L-3-HN, and Gly. The issue and discussion raised in the manuscript are very exciting, and most of them can be agreeable as a whole, but it is advisable that there remains some more deeper insight that need to be clarified for considering the mechanistic point of view behind this complex, unexplored reaction.

As a major point concerning the mechanism for EDMP formation, author must first clarify the stoichiometry of the chemoenzymatic process, in which 3 equivalents of L-Thr (C4) may be converted to one eq. of EDMP (C8), one eq. of glycine (C2), and two eq. of CO_2 . The designation presented in the Figure 1 is not clear as for the stoichiometry issue. It should be pointed out in my opinion that decarboxylation can be a significant part of the TDH catalysis because AKB has the structure of beta-keto-acid form, which spontaneously decomposes by releasing CO_2 off the molecule. Decarboxylation may occur before the dehydrogenated product is released from TDH because the enzyme has an Asp residue that is associated with the pyridine-N in the PLP prosthetic group. This is typically seen in aminotransferase and decarboxylation enzymes. Accordingly, the authors must clarify the discrete role of the two

enzymes contributing to produce 1 eq. EDMP from 3 eq. of L-Thr substrates.

Chemical reaction mechanism, besides the enzymatic catalysis, must also be discussed in detail. How two moles of aminoacetone can form pyrazines and what is the role of aldehydes in drawing the equilibrium in favor of forward direction and to make the 3-alkyl-2,5-dimethylpyrazines.

Minor points

P2. Line 29

We surmise that EDMP...

The choice of word 'surmise' is not convincing enough for usage in paper. Readers want conclusion based on lines of evidence rather than speculation.

P2. Line 31

Aminoacetone was supplied by TDH which converts L-Thr to aminoacetone via the unstable 2-amino-3-ketobutyrate.

This sentence is written in past passage, but the result does not represent the evidence of aminoacetone formation nor the detection of 2-amino-3-ketobutyrate. The authors must show the evidence if this sentence is the experimental results.

P4. Line 84.

The group stated that

The authors must select an appropriate word when quoting other paper. The word 'stated' does not convey information how scientifically sufficient evidence did the previous paper contained. Was it just a speculation or was it firmly established by lines of evidence?

P5. Line 94.

In this study, we attempted find another mechanism by which to synthesize pyrazines from L-Thr.

If the purpose of the present study was on finding a novel reaction mechanism, it requires the effort of kinetic studies for elucidating kinetic constants at every stage of reaction, but this paper even did not succeed to clarify discrete role of the two enzymes and chemical reaction scheme after the product is released from enzymes. To work on mechanistic issue, work on kinetics of each enzyme reaction and chemical reaction.

P5. Line 104 bifunctional

The reviewer can hardly agree to call this phenomena as 'bifunctionality of enzyme'. This can merely be an artificial reaction that occurred in the absence of required cofactor CoASH.

The author must characterize the affinity of the enzyme to CoASH to discuss whether this could happen under physiological conditions or not?

P5. Line 110

...pET28a-CnKBL, which cloned Cupriavidus ...

Is the plasmid subject and the verb is to clone? It is a researcher who clones a gene.

P6. Line 124

The culture was cultivated

The sentence is Redundant.

P6. Line 129

...was applied to a Ni²⁺ Sepharose column.

What is the volume of the column? Ambiguous description.

P7. Line 135.

Purity was confirmed by sodium dodecyl sulfate-polyacrylamide gel electrophoresis.

It must describe how proteins were stained on the gel after the electrophoresis. Ambiguous

description.

P11. Line 225

0.05 mM CnTDH

The concentration of the CnTDH must be rather high if this is true.

P13. Line 281

...this compound could be EDMP.

The author seems to be much less confident of EDMP that was produced in the enzyme reaction. In fact, this paper has very limited evidence that supports the EDMP being produced.

P14. Line 293

an unexpected carbon-carbon bond cleavage reaction.

There is not an unexpected C-C bond cleavage in the enzyme reaction. There is either one of the two C-C bond cleavages clearly expected according to PLP-dependent enzymes.

P15. Line 319

...and therefore we hypothesized that CnKBL has TA activity.

The TA activity of CnKBL depends on the absence or low concentration of CoASH according to the results in Figure 1 b-d. It is recommended to measure the K_m value for CoASH for the subject CnKBL, and discuss whether catalysis could promote aldorase reaction or acetyltransferase reaction under the physiological conditions.

P17-18

Crystallographic studies certainly suggested putative catalytic role of residues for aldorase reaction on PLP-bound Thr. How about the acetyltransferase mechanism? Was there an open cavity for CoA to be bound for accepting acetyl group near the active site?

P19, Line 416.

Give the yield of the variants produced in combination of diverse aldehydes.

In addition to responding to the reviewers comments, I corrected many sentences that were difficult to read. **All corrections in the manuscript are shown in red.**

- The abstract has been fully revised.
- EDMP biosynthesis in vivo and stable isotope tracing experiments were performed. Figures 2 and 3 have been added.
- Figure 7a) and Figure 8 was revised.
- Wordy and redundant phrases were removed.
- Gene names were italicized. For example, *tdh*, *kbl*.

Response to reviewer #1

We deeply appreciate the reviewer both for his/ her patience, valuable time and precise comments/ questions. Our answers were written in followings. Modification points were highlighted by red color in the manuscript. *The comments/questions are shown in italicized blue text.* **All corrections in the manuscript are shown in red.**

Major comments:

1. *Stoichiometry of the reaction should be address. For example, reaction yields of EDMP and other pyrazines, which could be produced from the reaction intermediate, aminoacetone, are required. The author's model of the entire reaction (Fig. 6) is hardly acceptable in the absence of these data.*

Response: To follow this comment, we attempted to determine stoichiometry of our proposed reaction process by quantifying concentration of produced EDMP. However, because of following technical hurdles, we cannot determine the stoichiometry accurately. Therefore, Figure 8 (Figure 6 in previous version) was revised.

- We cannot achieve quantification of Gly, CO₂, and unstable intermediates, such as (Figure 7a) produced by the reactions.
- Conversion rate for condensation of aminoacetone and acetaldehydes to EDMP is limited, in vivo (new experiment, Fig. 3).

Main purpose of this manuscript is that EDMP can be synthesized only from L-Thr by TDH and KBL. Determination of stoichiometry and production yield would be required to apply this reaction to synthesize alkylpyrazines at industrial scale; the reaction system is constructing for now. In this situation, we have to validate that EDMP was produced by these two enzymes (TDH and KBL) with other data. Based on the following results, we believe that our proposed mechanism is reasonable.

1a. By following the comment No.4 of reviewer #1, we can prove that the products of CnTDH and CnKBL are incorporated into EDMP from a shift of their *m/z* value (Fig. 2). All carbon and nitrogen atoms of EDMP derived from isotope-labeled L-Thr.

1b. Strain of *Cupriavidus necator* can generate EDMP by supplying L-Thr under oligotrophic condition (Fig. 3).

1c. Incorporation of aminoacetone produced by TDH into alkylpyrazines was reported by other researchers (Zhang et al., *AEM*, 2019, **85**, e01807-19, and Papenfort et al., *Nat. Chem. Biol.*, 2017, **13**, 551-557). Although they reported that alkylpyrazines were

produced from metabolites of L-Thr and glucose or amino acids (Zhang et al., *AEM*, 2019, **85**, e01807-19, and Papenfort et al., *Nat. Chem. Biol.*, 2017, **13**, 551-557), they did not report the stoichiometry because of above mentioned technical hurdles.

Simultaneously, we admit that the model represented in Fig. 6 in previous version cannot be validated only from currently available data. So to reflect this point, we modified the manuscript as follows.

(After) a. We revised the Figures 7a and Fig. 8 (6 in previous version) and added Figures 2 and 3.

b. The rough conversion rate of EDMP in vitro experiment: “As the concentration of CoA decreases, the unknown compound of 137.1 m/z increased (Fig. 1d, red, blue and green lines). Curiously, the production of Gly was still confirmed without CoA (Fig. 1c, green line). Besides, these reactants had a strong nutty aroma. Judging from the m/z of LC-MS and mass spectrum of GC-MS analyses (Fig. 1d and Supplementary Fig. 2), this compound was identified as EDMP. Estimates from the TIC showed that about 20-25% of the L-Thr was converted to EDMP at no or 100 μ M CoA concentrations..” (lines 309-315, p. 15)

c. Following sentences were added to show that the project to apply our proposed reaction system to synthesize alkylpyrazines: “The next challenge is to optimize the synthetic method to achieve maximum yields of alkylpyrazines utilizing L-Thr as substrates. We are now trying to construct the method by combining chemo-enzymatic reactions.” (Conclusion section, lines 512-515 p. 24)

2. *Is the TA activity high enough to consider its primary role as TA rather than KBL under physiological conditions? Kinetic analysis of KBL may help understanding this.*

Response: Accordingly, we determined enzyme kinetics parameters for KBL activity. To describe this point, we added the following modifications in the manuscript.

(After) a. Kinetics parameters of KBL activity for CnKBL were represented in Supplementary Table 4.

b. we added the following sentences to compare TA and KBL activity: “

The turnover rate for the TA activity of CnKBL was comparable to that of *Streptomyces coelicolor* TA³⁷, but was much lower than that of *E. coli* TA (EcTA)³⁸. The catalytic efficiency (k_{cat}/K_m) of CnKBL toward L-Thr was 40-fold lower than that of EcTA. Next, the KBL activity was verified. Due to the instability of AKB substrate and the difficulty in detecting products, Mukherjee *et al.* uses glycine as a substrate and detect the formation of the thiol group using Ellman's reagent (Supplementary Fig. 7)³⁶. Therefore, the kinetic parameters of condensation activity in CnKBL were determined and compared with those of EcKBL³⁶. The k_{cat} , K_m and catalytic efficiency values of CnKBL for Gly were quite comparable to those of EcKBL. The apparent KBL activity of CnKBL is very superior to the TA activity of CnKBL. However, the K_m value of CnKBL was 4.1 times higher than that of EcKBL, suggesting that its activity is susceptible to the concentration of CoA (Supplementary Fig. 8 and Talbe4). In fact, in

low nutrient conditions, the concentrations of CoA and acetyl-CoA in animal and bacterial cells have been reported to be reduced to 1.3-20 μmol ^{43, 44}. These results are consistent with the EDMP production *in vivo* and discussions mentioned above. (lines 383-398, pp.18-19). Experimental procedure was also added.

3. *Physiological role of the CnTDH-CnKBL operon is poorly supported by the data. Production of EDMP by Cupriavidus bacterium demonstrates conclusive evidence.*

Response: Accordingly, we try to represent that living *C. necator* cell can generate EDMP only from L-Thr. Production of EDMP by *C. necator* was analyzed by GC-MS under eutrophic (NBRC medium, Fig. 3, black line) and oligotrophic condition (M9 medium, Fig. 3, red line), respectively. To describe the results, we added the following sentences in the manuscript: “Besides, utilizing L-Thr as a substrate, EDMP could be synthesized under physiological conditions in the strain of *C. necator*. GC-MS analysis indicated that there is a peak corresponding with EDMP after 3 days of cultivation under M9 medium containing L-Thr but not in NBRC medium (Fig. 3). This suggested that EDMP would be synthesized under oligotrophic conditions in the strain.” (lines 338-343, p. 16) Experimental procedure, Fig. 3 and some related sentences have been corrected. OK

4. *I recommend CnTDH-CnKBL reactions containing isotope-labeled threonine to clarify the 3-ethyl residue of EDMP originating from acetaldehyde.*

Response: Accordingly, we performed the CnTDH-CnKBL reactions utilizing isotope-labeled threonine, and confirmed that products of CnTDH and CnKBL are incorporated into EDMP. To describe this point, we added the following sentences in the manuscript.

(After): Next, we used $[\text{U-}^{13}\text{C},^{15}\text{N}]$ -L-Thr to check whether the supplied L-Thr is incorporated into EDMP by CnTDH and CnKBL. Increased units of mass (e.g. 10 unit, m/z 145.1-135.1) indicates that all carbon and nitrogen atoms of EDMP derived from isotope-labeled L-Thr. (lines 323-326, p. 15) Experimental procedure, Fig. 2 and some related sentences have been corrected.

Minor comments:

1. *Line 110, Origin of the plasmid, and the bacterium should be provided.*

Response: Accordingly, we added information about origin of the plasmid, and the bacterium.

(After): Dried cells of *C. necator* (NBRC 102504) was purchased from NBRC (Biological Resource Center, NITE). Gene for *cnkbl* on the genome of *C. necator* was amplified by PCR using primers (Supplementary Table 1). The amplified gene was subcloned into pET28a vector cleaved by *NcoI/XhoI*. The prepared pET28a-cnkbl plasmids were utilized in the following experiment. (lines 110-114, p. 6)

2. *Line 310, Missing reference.*

Response: Accordingly, we added two references in the manuscript.

(After): ... is a product of TDH catalysis^{21,36}. (line 352, p. 17)

3. *Line 356, changes*

Response: Accordingly, we modified the tense of “changes” as follows.

(After): changed (line 405, p. 19)

4. *Please note molar relationship of the chemicals.*

Response: Accordingly, we added the information on molar relationships of the chemical and added the following sentences in the manuscript.

(After): Here, we predicted that the condensation of two aminoacetones and acetaldehyde would progress by following the reaction scheme as shown in Fig. 7a. Aminoacetone supplied by CnTDH would be condensed and dehydrated to 2,5-dimethyl-3,6-dihydropyrazine (DHP); the production of DHP has been already reported by other groups^{5, 16}. Through tautomerization, additional reaction of DHP and acetaldehyde, and subsequent dehydration reaction, EDMP would be produced. (lines 472-478, p. 22).

Response to Reviewer #2

We appreciate you to read our manuscript and give kind comments. Modification points were highlighted by red color in the manuscript. The responses to the comments shown as followings. *The comments/questions are shown in italicized blue text. All corrections in the manuscript are shown in red.*

Major points

As a major point concerning the mechanism for EDMP formation, author must first clarify the stoichiometry of the chemoenzymatic process, in which 3 equivalents of L-Thr (C4) may be converted to one eq. of EDMP (C8), one eq. of glycine (C2), and two eq. of CO₂. The designation presented in the Figure 1 is not clear as for the stoichiometry issue.

Reviewer 1 also pointed out for the concern about determination of stoichiometry of our proposed reaction. For this point, we wrote the response as shown in answer to comment 1 of reviewer 1 (Please see the answer). In summary, because of several of technical hurdles, it is hard to determine the stoichiometry accurately. However, substantial EDMP produced in vitro and in viro under physiological concentration of CoA. Thus, we modified the manuscript with referring to the already published paper which reported production of alkylpyrazines utilizing metabolites of L-Thr and glucose or other amino acids but not to mention the stoichiometry (Zhang et al., *AEM*, 2019, **85**, e01807-19, and Papenfort et al., *Nat. Chem. Biol.*, 2017, **13**, 551-557).

It should be pointed out in my opinion that decarboxylation can be a significant part of the TDH catalysis because AKB has the structure of beta-keto-acid form, which spontaneously decomposes by releasing CO₂ off the molecule. Decarboxylation may occur before the dehydrogenated product is released from TDH because the enzyme has an Asp residue that is associated with the pyridine-N in the PLP prosthetic group. This is typically seen in aminotransferase and decarboxylation enzymes. Accordingly, the authors must clarify the discrete role of the two enzymes contributing to produce 1 eq. EDMP from 3 eq. of L-Thr substrates.

For this comment, we firstly apologize for insufficient description about TDH. TDH is NAD⁺ dependent enzyme (and not PLP-dependent enzyme) catalyzing dehydrogenation of side chain hydroxyl group of L-Thr to AKB. TDH catalyzes only dehydrogenation reaction. The product, AKB, is rapidly decarboxylated to aminoacetone and CO₂ as already reported in previous studies (Mukherjee et al., *J. Biol. Chem.*, 1987, **262**, 30, 14441-7, and Schmidt et al., *Biochemistry*, 2001, **40**, 17, 5151-60). Transfer mechanism of AKB from TDH to KBL is essential process to produce the Acetyl-CoA, but the mechanism remains unsolved questions for now.

Summarizing previous researches and our suggested results for TDHs and KBLs, we predicted the roles of these enzymes as follows: TDH converts L-Thr to AKB, and AKB is metabolize to Gly by KBL in usual condition. However, when they are poorly

nourished and the concentration of CoA is low, unstable AKB is decarboxylated to aminoacetone, and KBL converts L-Thr to acetaldehyde and Glycine. To complement this point and clarify the roles of the enzymes, we added following sentences in the manuscript.

(After) TDH converts L-Thr to AKB, and AKB is metabolized to Gly by acetyltransferase activity of KBL in usual conditions (Fig. 8). However, in poor nutrition and low CoA concentration, unstable AKB is gradually decarboxylated to aminoacetone, and lyase activity of KBL decomposes L-Thr to acetaldehyde and Glycine (Fig. 7a and 8). (Conclusion section, lines 508-512, p. 24)

Chemical reaction mechanism, besides the enzymatic catalysis, must also be discussed in detail. How two moles of aminoacetone can form pyrazines and what is the role of aldehydes in drawing the equilibrium in favorer of forward direction and to make the 3-alkyl-2,5-dimethylpyrazines.

Response: Accordingly, we revised the sentences and Fig 7a.

(After): Here, we predicted that the condensation of two aminoacetones and acetaldehyde would progress by following the reaction scheme as shown in Fig. 7a. Aminoacetone supplied by CnTDH would be condensed and dehydrated to 2,5-dimethyl-3,6-dihydropyrazine (DHP); the production of DHP has been already reported by other groups^{5, 16}. Through tautomerization, additional reaction of DHP and acetaldehyde, and subsequent dehydration reaction, EDMP would be produced. (lines 472-478, p. 22). Fig. 7a and some related sentences have been corrected.

Minor points

1. P2. Line 29, *We surmise that EDMP*

The choice of word 'surmise' is not convincing enough for usage in paper. Readers want conclusion based on lines of evidence rather than speculation.

Response: The abstract has been fully revised.

2. P2. Line 31, *Aminoacetone was supplied by TDH which converts L-Thr to aminoacetone via unstable 2-amino-3-ketobutyrate.*

This sentence is written in past passage, but the result does not represent the evidence of aminoacetone formation nor the detection of 2-amino-3-ketobutyrate.

The authors must show the evidence if this sentence is experimental results.

Response: Some groups reported aminoacetone was supplied by TDH (Mukherjee et al., *J. Biol. Chem.*, 1987, **262**, 30, 14441-7, and Schmidt et al., *Biochemistry*, 2001, **40**, 17, 5151-60). So we should add these references in the manuscript to describe this point. However, we cannot add the references at page 2, because the sentence is placed in Abstract section. As an alternative, references were added to page 17.

3. P4. Line 84, *The group stated that*

The author must select an appropriate word when quoting other paper. The word

'stated' does not convey information how scientifically sufficient evidence did the previous paper contained. Was it just a speculation or was it firmly established by lines of evidence?

Response: Accordingly, we modified the word of "stated" as following.

(After): reported (line 83, p. 4)

4. *P5, Line 94, In this study, we attempted find another mechanism by which to synthesize pyrazines from L-Thr*

If the purpose of the present study was on finding a novel reaction mechanism, it requires the effort of kinetic studies for elucidating kinetic constants at every stage of reaction, but this paper even did not succeed to clarify discrete role of the two enzymes and chemical reaction scheme after the product released from enzymes. To work on mechanistic issues, work on kinetics of each enzyme reaction and chemical reaction.

Response: In this study, we try to find synthetic pathway of alkylpyrazine in bacteria utilizing only L-Thr as substrates. As pointed out by this comment, we agree that the sentence in previous manuscript cannot reflect the purpose of this study correctly. To amend this point, we modified the manuscript as followings.

(After) a. we added the sentence: we attempted to find synthetic pathway in bacteria which can produce alkylpyrazine only from L-Thr (lines 93-94, p. 5).

b. Enzyme kinetics parameters for KBL activity of CnKBL was determined (Supplementary Table 4).

c. We added the experimental data supporting that strain of *C. necator* can synthesize EDMP only from L-Thr under oligotrophic condition (p.16 and Fig. 3)

5. *P5, Line 104, bifunctional*

The reviewer can hardly agree to this phenomena as 'bifunctionality of enzyme'. This can merely be an artificial reaction that occurred in the absence of required cofactor CoASH. The author must characterize the affinity of the enzyme to CoASH to discuss whether this could happen under physiological conditions or not

Response: As pointed out, we agree that specificity constant of TA activity is lower than that of KBL activity in CnKBL. However, enzymatic properties of KBL suggest that acetyltransferase activity gradually shifts to aldolase activity in physiological condition. Simultaneously, we can obtain experimental data *in vivo* suggesting that *C. necator* and related microorganisms can synthesize EDMP under physiological conditions. To reflect these points, we amended the manuscript as follows.

(After) a. "We predicted that CnKBL catalyzes not only acetyltransferase reaction (EC 2.3.1.29) which is broadly recognized by previous studies but also another reaction. Based on other PLP-dependent enzymes, CnKBL may catalyze carbon-carbon bond cleavage reaction (EC 4.1.X.X) as a side reaction." (lines 328-331, p.16)

b. We added the sentences to support that EDMP can be synthesized under physiological condition as described above: "Besides, utilizing L-Thr as a substrate,

EDMP could be synthesized under physiological conditions in the strain of *C. necator*. GC-MS analysis indicated that there is a peak corresponding with EDMP after 3 days of cultivation under M9 medium containing L-Thr but not in NBRC medium (Fig. 3). This suggested that EDMP would be synthesized under oligotrophic conditions in the strain.” (line 338-343, p.16)

c. We added enzyme kinetics parameters of KBL activity in CnKBL (lines 383-398, pp.18-19 and Supplementary Table 4).

6. P5, Line 110

pET28a-CnKBL, which cloned Cupriavidus

Is the plasmid subject and the verb is to clone? It is a researcher who clones a gene.

Response: Accordingly, we modified the sentence.

(After): Dried cells of *C. necator* (NBRC 102504) was purchased from NBRC (Biological Resource Center, NITE). Gene for *cnkbl* on the genome of *C. necator* was amplified by PCR using primers (Supplementary Table 1). The amplified gene was subcloned into pET28a vector cleaved by *NcoI/XhoI*. The prepared pET28a-*cnkbl* plasmids were utilized in the following experiment. (lines 110-114, p.6)

7. P6, Line 124

The culture was cultivated

The sentence is Redundant.

Response: Accordingly, we modified the sentence as followings.

(After) pET28a-*cnkbl* in BL21(DE3) was cultivated overnight at 37 °C in 5 mL of Luria-Bertani (LB) medium containing 30 µg/mL kanamycin. (lines 123-124, p.6)

8. P6, Line 129

... was applied to a Ni²⁺-Sepharose column.

What is the volume of the column? Ambiguous description.

Response: Accordingly, we added the volume of the column as following.

(After): ... a 5 mL of Ni²⁺-Sepharose column. (line 133, p.7)

9. P7, Line 135

Purity was confirmed by sodium dodecyl sulfate-polyacrylamide gel electrophoresis.

It must describe how proteins were stained on the gel after the electrophoresis. Ambiguous description.

Response: Accordingly, we added the method of staining the gel as following.

(After): The gel was stained with coomassie brilliant blue R-250 (Wako). (lines 140-141, p.7)

10. P11, Line 225

0.05 mM CnTDH

The concentration of the CnTDH must be rather high if this is true.

The reaction was performed under the condition containing total 0.5 mL of assay buffer. So we believe the concentration of the CnTDH is not so high. To complement this point, we added the following modification.

(After) The chemoenzymatic reaction was performed using 0.5 mL of assay buffer F...
(line 273, p.13)

11.P13, Line 281

this compound could be EDMP

The author seems to be much less confident of EDMP that was produced in the enzyme reaction. In fact, this paper has very limited evidence that supports the EDMP being produced.

In this study, we assigned the production of EDMP by m/z value of LC-MS analysis and library search of fragment ions of the compounds obtained by GC-MS analysis. To complement this point, we amended the sentence as following.

(After) Besides, these reactants had a strong nutty aroma. Judging from the m/z of LC-MS and mass spectrum of GC-MS analyses (Fig. 1d and Supplementary Fig. 2), this compound was identified as EDMP. (lines 311-314, p.15)

12.P14, Line 293

An unexpected carbon-carbon bond cleavage reaction

There is not an unexpected C-C bond cleavage in the enzyme reaction. There is either one of the two C-C bond cleavage clearly expected according to PLP-dependent enzymes.

As pointed out by this comment, we admit that the description is inappropriate. To amend this point, we modified the sentence as following.

(After) We predicted that CnKBL catalyzes not only acetyltransferase reaction (EC 2.3.1.29) which is broadly recognized by previous studies but also another reaction. Based on other PLP-dependent enzymes, CnKBL may catalyze carbon-carbon bond cleavage reaction (EC 4.1.X.X) as a side reaction. (line 328-331, p.16)

13.P15, Line 319

And therefore we hypothesized that CnKBL has TA activity.

The TA activity of CnKBL depends on the absence or low concentration of CoASH according to the results in Figure 1 b-d. It is recommended to measure the K_m value for CoASH for the subject CnKBL, and discuss whether catalysis could promote aldolase reaction or acetyltransferase reaction under physiological condition.

Response: Accordingly, we measured the K_m value for Acetyl-CoA in lieu of CoA. This is because we cannot utilize fragile substrates, AKB. To answer this comment, we added the following descriptions in the manuscript.

(After) The catalytic efficiency (k_{cat}/K_m) of CnKBL toward L-Thr was 40-fold lower than that of EcTA. Next, the KBL activity was verified. Due to the instability of AKB substrate and the difficulty in detecting products, Mukherjee *et al.* uses glycine as a substrate and detect the formation of the thiol group using Ellman's reagent (Supplementary Fig. 7)³⁶. Therefore, the kinetic parameters of condensation activity in CnKBL were determined and compared with those of EckBL³⁶. The k_{cat} , K_m and catalytic efficiency values of CnKBL for Gly were quite comparable to those of EckBL. The apparent KBL activity of CnKBL is very superior to the TA activity of CnKBL. However, the K_m value of CnKBL was 4.1 times higher than that of EckBL, suggesting that its activity is susceptible to the concentration of CoA (Supplementary Fig. 8 and Talbe4). In fact, in low nutrient conditions, the concentrations of CoA and acetyl-CoA in animal and bacterial cells have been reported to be reduced to 1.3-20 μmol ^{43, 44}. These results are consistent with the EDMP production *in vivo* and discussions mentioned above. (lines 385-398, pp. 18-19)

14. P17-18

Crystallographic studies certainly suggested putative catalytic role of residues for aldolase reaction on PLP-bound Thr. How about the acetyltransferase mechanism? Was there an open cavity for CoA to be bound for accepting acetyl group near the active site?

Response: Accordingly, we added the following sentences in the manuscript.

(After): Here, the reaction mechanism of KBL activity for CnKBL is identical to that of the already reported EckBL^{21, 39}. As supporting data, the sequence identity between EckBL and CnKBL is high (61%), and active site residues, S187 (S185 in EckBL) and H215 (H213 in EckBL), are conserved to each other (Supplementary Fig. 9). Furthermore, CnKBL has a cavity to bind CoA to the active site as well as EckBL²¹. (lines 407-412, pp. 19)

15. P19, Line 416

Give the yield of the variants produced in combination of diverse aldehydes.

As pointed out by this comment, we agree that it is better to estimate product yield of each alkylpyrazines. However, we cannot optimize reaction methods to achieve in maximum yield of alkylpyrazines. The product yield should be determined after completion of the optimization because we may underestimate the product yield in the current situation. To mention this point, we added the following description in the manuscript.

(After) The next challenge is to optimize the synthetic method to achieve maximum yields of alkylpyrazines utilizing L-Thr as substrates. We are now trying to construct the method by combining chemo-enzymatic reactions. (conclusion section, line 512-516, p.24)

Reviewers' comments:

Reviewer #1 (Remarks to the Author):

Unfortunately, this version still lacks stoichiometry of the reaction, and does not meet the basic requirement of chemistry. I do not recommend this paper for the publication in this journal.

Reviewer #2 (Remarks to the Author):

Thank you for considering comments and suggestions for revising the manuscript. It is unfortunate that stoichiometric analysis was hard to perform, but this may be spared for your continuing research.

Even though I am satisfied with the scientific contents in the revised manuscript, I still feel stress while reading the text. I presume it is for the reason of Japan-english style writing. Sentences and compositions are awkward, and I wish the manuscript be submitted to some English-editing service, hopefully, that arrange two editing specialists, one for science and the other for style-writing. Then your paper will be smooth for every native readers, and highly evaluated.

sincerely

Reviewer #3 (Remarks to the Author):

I read through the manuscript and then looked at the review comments and rebuttal. The authors do claim in their abstract that substantial amounts of EDMP can be synthesized, yet can't quantify it, and I'm not sure other than molecular mass they have evidence it exists e.g. no standard used in LCMS/GCMS, no NMR data. The papers they refer to as also having technical limitations preventing stoichiometric analysis also have a very different focus. Considering that this study relates to a novel enzyme with novel activities and mechanisms, and the authors admit that they can't prove their mechanism with the data they have, I don't think that the study is complete enough for publication in this journal.

In addition to responding to the reviewers' comments, we corrected some sentences that were difficult to read. Please note that the updated manuscript includes the following corrections.

- NMR analysis of chemoenzymatically produced EDMP was performed. Supplementary Figure 3 has been added.
- The yield of EDMP produced by CnTDH and CnKBL from L-Thr was determined. Supplementary Figure 4 have been added.
- Comparison of EDMP and DMP productions under non-enzymatic and enzymatic conditions was added in the result & discussion section. Figure 13, Figure 14 and Supplementary Table 6 have been added.

As most people know, pyrazine is an aroma component produced by heating, as is typical in coffee and peanuts. However, in the chemoenzymatic reaction that we found in this study, the rate of EDMP formation clearly decreases when the reaction condition exceeds a medium temperature. Supplementary Table 6, added in this revision, clearly illustrate this point. This may be evidence that some pyrazines can be synthesized under biological conditions and can express physiological functions.

Response to reviewer #1

We appreciate you to read our manuscript and give kind comments. Modification points were highlighted by red color in the manuscript. The responses to the comments shown as followings. *The comments/questions are shown in italicized blue text.* **All corrections in the manuscript are shown in red.**

Comments:

- 1. Unfortunately, this version still lacks stoichiometry of the reaction, and does not meet the basic requirement of chemistry.*

Response: As shown in Supplementary Table 6 and new section named "Yields of EDMP and DMP under chemical or chemoenzymatic reactions." The condensation reaction between aminoacetone and acetaldehyde generate EDMP and DMP. This suggested that we cannot indicate accurate stoichiometry with experiment because the DMP would be formed. In this situation, assignment of reaction intermediate would be an alternative to support the proposed reaction mechanism, here, DHP is some instable compound and therefore, we could not assign DHP directly. Real-time NMR experiments were also tried, but as in previous reports, the intermediates could not be detected. However, table 6 clearly shows that the intermediates is slowly converted to DMP in a maximum yield of about 10%. In the presence of aldehyde, EDMP is produced via nucleophilic addition reaction, with a maximum yield of 16.2% under physiological temperature. The fact that the yield exceeded 20% in the chemoenzymatic reaction may be explained by the similarity of the optimal reaction temperature between the enzymatic and chemical reactions and the accelerating effect of TDH and KBL on EDMP synthesis. To illustrate these points, we added the following description to the manuscript.

(After): ... and NMR analysis (Supplementary Fig. 3), this compound was identified as EDMP. HPLC analysis showed that the yield of EDMP from L-Thr was approximately 4%. The yield of EDMP was calculated from Supplementary Fig. 4. (lines 320–323, P15–16)

Comparison of EDMP production under chemical and chemoenzymatic reactions

Structural and functional analysis of CnTDH and CnKBL suggested that EDMP could be synthesized only from L-Thr with chemoenzymatic reaction. Aminoacetone and acetaldehyde, precursors to generate EDMP, are supplied by enzymatic reactions, and chemical reactions progress their condensation. The next challenge is to optimize the reaction conditions to maximize the amount of EDMP produced. It is expected that reaction temperature and timing of addition of precursor would affect the rate of EDMP production.

Firstly, we estimated the optimal temperature for the maximum synthesis of EDMP by changing the reaction temperature (Entry 1–4 in Supplementary Table 6). The yield of EDMP was as high as 16.2% under the chemical reaction condition of 30 °C. Notably, this yield gradually decreased as the temperature increased. In the Maillard reaction, pyrazine is generated at high temperatures^{9, 10}. The fact that the chemical condensation reaction of aminoacetone and acetaldehyde proceeds under such moderate conditions may have implications for the identification of pyrazine compounds *in vivo* (Fig. 3).

The synthesis mechanism of EDMP is shown in Fig. 7a. First, as is well known, aminoacetone is supplied from L-Thr by TDH. Next, the two aminoacetones condense to 2,5-dimethyl-3,6-dihydropyrazine (DHP). Finally, DHP tautomerizes and undergoes nucleophilic addition to acetaldehyde to synthesize EDMP. Other research groups reported that the DHP is easily oxidized under mild condition to form 2,5-dimethylpyrazine (DMP)^{42, 43}, suggesting that immediate reaction of DHP with acetaldehyde increases the production of EDMP. To confirm this point, we compared production rates of EDMP and DMP by changing the timing of acetaldehyde addition (Supplementary Table 6). As expected, yield of EDMP is maximized at the condition which mixed acetaldehyde and aminoacetone simultaneously (16.2%, entry 5 in Supplementary Table 6). By delaying the timing of addition, the production rate gradually decreased. When acetaldehyde was added after 12 hours of pre-reaction, the EDMP production rate was minimal (Entry 9 in Supplementary Table 6). On the other hand, the opposite tendency was observed for the production rate of DMP (Entry 5–9 in Supplementary Table 6 and Supplementary Fig. 13 and 14).

Considering the time-dependent changes in the production rates of EDMP and DMP, it was found that the reaction intermediate, DHP, was stable for a certain time and that the reaction of DHP and acetaldehyde at physiological temperature was

important for the chemoenzymatic synthesis of EDMP. When the precursor was supplied from L-Thr by the enzymes, the yield of EDMP was increased up to 20.2%.

(lines 474–511, P22–24)

Experimental procedures rerated to Supplementary Fig. 3 and 4, and some related sentences have been corrected.

Response to reviewer #2

We deeply appreciate the reviewer both for his/ her patience, valuable time and precise comments/ questions. Our answers were written in followings. Modification points were highlighted by red color in the manuscript. *The comments/questions are shown in italicized blue text.* All corrections in the manuscript are shown in red.

Comments:

- 1. Even though I am satisfied with the scientific contents in the revised manuscript, I still feel stress while reading the text. I presume it is for the reason of Japan-english style writing. Sentences and compositions are awkward, and I wish the manuscript be submitted to some English-editing service, hopefully, that arrange two editing specialists, one for science and the other for style-writing. Then your paper will be smooth for every native readers, and highly evaluated.*

Response: Accordingly, we submitted the manuscript to English-editing service.

Response to reviewer #3

Thank you for giving some critical comments with carefully reading of our manuscript. Our answers were written in followings. Modification points were highlighted by red color in the manuscript. *The comments/questions are shown in italicized blue text. All corrections in the manuscript are shown in red.*

Comments:

1. *The authors do claim in their abstract that substantial amounts of EDMP can be synthesized, yet can't quantify it.*

Response: We calculated the yield of EDMP produced by CnTDH and CnKBL from L-Thr. To describe this point, we added the following description in the manuscript.

(After): ... HPLC analysis showed that the yield of EDMP from L-Thr was approximately 4%. The yield of EDMP was calculated from Supplementary Fig. 4. (lines 321-323, P15–16)

As expected, yield of EDMP is maximized at the condition which mixed acetaldehyde and aminoacetone simultaneously (16.2%, entry 5 in Supplementary Table 6).

(lines 498-500, P24)

When the precursor was supplied from L-Thr by the enzymes, the yield of EDMP was increased up to 20.2%.

(lines 509-511, P24)

Experimental procedure related to Supplementary Fig. 4 and some related sentences have been corrected.

2. *and I'm not sure other than molecular mass they have evidence it exists e.g. no standard used in LCMS/GCMS, no NMR data.*

Response: We added the GC-MS data using EDMP standard (Supplementary Fig. 2) and NMR data of EDMP produced by CnTDH and CnKBL from L-Thr (Supplementary Fig. 3). To describe this point, we added the following description in the manuscript:

(After): Judging from the m/z of LC-MS (Fig. 1d), mass spectrum of GC-MS (Supplementary Fig. 2), and NMR analysis (Supplementary Fig. 3), this compound was identified as EDMP. (lines 319-321, P15)

Experimental procedures related to Supplementary Fig. 2 and 3, and some related sentences have been corrected.

3. *The papers they refer to as also having technical limitations preventing stoichiometric analysis also have a very different focus.*

Response: Reviewer 1 also pointed out for the concern about determination of stoichiometry of our proposed reaction. For this point, we wrote the response as shown in answer to comment 1 of reviewer 1 (Please see the answer). In summary, we estimated the yield of EDMP.

REVIEWERS' COMMENTS:

Reviewer #1 (Remarks to the Author):

The authors added new experiments for conversion yields of EDMP (Supplementary table 6), which lost impact of the paper.

They shows only 4% and minor contribution of TDH/KBL, and no explanation in major reaction product originated from around 80% of Thr. I feel very small advance of the use of TDH/KBL for applied/process chemistry for pyrazine production.

Reviewers' comments:

Reviewer #1 (Remarks to the Author):

This paper demonstrates that the CnTDH/CnKBL couple convert L-threonine to EDMP. Threonine aldolase (TA) activity of CnKBL participates in the conversion reaction while Marcus et al. have already reported this activity of Escherichia coli KBL (BBA, 1993, 1164, 299-304), which minimizes the impact of this paper. The chemoenzymatic condensation of aldehydes and threonine/aminoacetone to alkylpyrazines may be applicable to pharmaceutical and flavor industries.

The proposed mechanism synthesizing EDMP may be novel. However, I don't think that the results are conclusive enough to show the author's claim, and cannot recommend this article for publication at this stage.

- 1) Stoichiometry of the reaction should be address. For example, reaction yields of EDMP and other pyrazines, which could be produced from the reaction intermediate, aminoacetone, are required. The author's model of the entire reaction (Fig. 6) is hardly acceptable in the absence of these data.
- 2) Is the TA activity high enough to consider its primary role as TA rather than KBL under physiological conditions? Kinetic analysis of KBL may help understanding this.
- 3) Physiological role of the CnTDH-CnKBL operon is poorly supported by the data. Production of EDMP by Cupriavidus bacterium demonstrates conclusive evidence.
- 4) I recommend CnTDH-CnKBL reactions containing isotope-labeled threonine to clarify the 3-ethyl residue of EDMP originating from acetaldehyde.

Minor comment:

Line 110, Origin of the plasmid, and the bacterium should be provided.

Line 310, Missing reference.

Line 356, changes

Figure 5a, Please note molar relationship of the chemicals.

Reviewer #2 (Remarks to the Author):

The present study reports a novel chemo-enzymatic reaction utilizing L-threonine-3-dehydrogenase (TDH) and 2-amino-3-ketobutyrate CoA ligase (KBL) as catalysts and L-Thr as a sole substrate can produce 3-ethyl-2,5-dimethylpyrazine (EDMP). The authors propose that the reaction appears to proceed by delivering aminoacetone ($\text{CH}_3\text{COCH}_2\text{-NH}_2$) and (acetaldehyde + glycine) in the absence of CoA. The authors also reported the formation of EDMP derivatives that differ in the side chain

group, which derives from aldehyde substrates added to the chemoenzymatic reactions. The authors also reported crystallographic evidences of enzymatic mechanisms for recognizing PLP-bound L-Thr, L-3-HN, and Gly. The issue and discussion raised in the manuscript are very exciting, and most of them can be agreeable as a whole, but it is advisable that there remains some more deeper insight that need to be clarified for considering the mechanistic point of view behind this complex, unexplored reaction.

As a major point concerning the mechanism for EDMP formation, author must first clarify the stoichiometry of the chemoenzymatic process, in which 3 equivalents of L-Thr (C4) may be converted to one eq. of EDMP (C8), one eq. of glycine (C2), and two eq. of CO₂. The designation presented in the Figure 1 is not clear as for the stoichiometry issue. It should be pointed out in my opinion that decarboxylation can be a significant part of the TDH catalysis because AKB has the structure of beta-keto-acid form, which spontaneously decomposes by releasing CO₂ off the molecule. Decarboxylation may occur before the dehydrogenated product is released from TDH because the enzyme has an Asp residue that is associated with the pyridine-N in the PLP prosthetic group. This is typically seen in aminotransferase and decarboxylation enzymes. Accordingly, the authors must clarify the discrete role of the two enzymes contributing to produce 1 eq. EDMP from 3 eq. of L-Thr substrates.

Chemical reaction mechanism, besides the enzymatic catalysis, must also be discussed in detail. How two moles of aminoacetone can form pyrazines and what is the role of aldehydes in drawing the equilibrium in favorer of forward direction and to make the 3-alkyl-2,5-dimethylpyrazines.

Minor points

P2. Line 29

We surmise that EDMP...

The choice of word 'surmise' is not convincing enough for usage in paper. Readers want conclusion based on lines of evidence rather than speculation.

P2. Line 31

Aminoacetone was supplied by TDH which converts L-Thr to aminoacetone via the unstable 2-amino-3-ketobutyrate.

This sentence is written in past passage, but the result does not represent the evidence of aminoacetone formation nor the detection of 2-amino-3-ketobutyrate. The authors must show the evidence if this sentence is the experimental results.

P4. Line 84.

The group stated that

The authors must select an appropriate word when quoting other paper. The word 'stated' does not convey information how scientifically sufficient evidence did the previous paper contained. Was it just a speculation or was it firmly established by lines of evidence?

P5. Line 94.

In this study, we attempted find another mechanism by which to synthesize pyrazines from L-Thr. If the purpose of the present study was on finding a novel reaction mechanism, it requires the effort of kinetic studies for elucidating kinetic constants at every stage of reaction, but this paper even did not succeed to clarify discrete role of the two enzymes and chemical reaction scheme after the product is released from enzymes. To work on mechanistic issue, work on kinetics of each enzyme reaction and chemical reaction.

P5. Line 104 bifunctional

The reviewer can hardly agree to call this phenomena as 'bifunctionality of enzyme'. This can merely be an artificial reaction that occurred in the absence of required cofactor CoASH. The author must characterize the affinity of the enzyme to CoASH to discuss whether this could happen under physiological conditions or not?

P5. Line 110

...pET28a-CnKBL, which cloned Cupriavidus ...

Is the plasmid subject and the verb is to clone? It is a researcher who clones a gene.

P6. Line 124

The culture was cultivated

The sentence is Redundant.

P6. Line 129

...was applied to a Ni² Sepharose column.

What is the volume of the column? Ambiguous description.

P7. Line 135.

Purity was confirmed by sodium dodecyl sulfate-polyacrylamide gel electrophoresis.

It must describe how proteins were stained on the gel after the electrophoresis. Ambiguous description.

P11. Line 225

0.05 mM CnTDH

The concentration of the CnTDH must be rather high if this is true.

P13. Line281

...this compound could be EDMP.

The author seems to be much less confident of EDMP that was produced in the enzyme reaction. In fact, this paper has very limited evidence that supports the EDMP being produced.

P14. Line293

an unexpected carbon-carbon bond cleavage reaction.

There is not an unexpected C-C bond cleavage in the enzyme reaction. There is either one of the two C-C bond cleavages clearly expected according to PLP-dependent enzymes.

P15. Line 319

...and therefore we hypothesized that CnKBL has TA activity.

The TA activity of CnKBL depends on the absence or low concentration of CoASH according to the results in Figure 1 b-d. It is recommended to measure the K_m value for CoASH for the subject CnKBL, and discuss whether catalysis could promote aldorase reaction or acetyltransferase reaction under the physiological conditions.

P17-18

Crystallographic studies certainly suggested putative catalytic role of residues for aldorase reaction on PLP-bound Thr. How about the acetyltransferase mechanism? Was there an open cavity for CoA to be bound for accepting acetyl group near the active site?

P19, Line416.

Give the yield of the variants produced in combination of diverse aldehydes.

In addition to responding to the reviewer's comments, I corrected many sentences that were difficult to read. **All corrections in the manuscript are shown in red.**

- The abstract has been fully revised.
- EDMP biosynthesis in vivo and stable isotope tracing experiments were performed. Figures 2 and 3 have been added.
- Figure 7a) and Figure 8 was revised.
- Wordy and redundant phrases were removed.
- Gene names were italicized. For example, *tdh*, *kbl*.

Response to reviewer #1

We deeply appreciate the reviewer both for his/ her patience, valuable time and precise comments/questions. Our answers were written in followings. Modification points were highlighted by red color in the manuscript. *The comments/questions are shown in italicized blue text.* **All corrections in the manuscript are shown in red.**

Major comments:

1. *Stoichiometry of the reaction should be address. For example, reaction yields of EDMP and other pyrazines, which could be produced from the reaction intermediate, aminoacetone, are required. The author's model of the entire reaction (Fig. 6) is hardly acceptable in the absence of these data.*

Response: To follow this comment, we attempted to determine stoichiometry of our proposed reaction process by quantifying concentration of produced EDMP. However, because of following technical hurdles, we cannot determine the stoichiometry accurately. Therefore, Figure 8 (Figure 6 in previous version) was revised.

- We cannot achieve quantification of Gly, CO₂, and unstable intermediates, such as (Figure 7a) produced by the reactions.
- Conversion rate for condensation of aminoacetone and acetaldehydes to EDMP is limited, in vivo (new experiment, Fig. 3).

Main purpose of this manuscript is that EDMP can be synthesized only from L-Thr by TDH and KBL. Determination of stoichiometry and production yield would be required to apply this reaction to synthesize alkylpyrazines at industrial scale; the reaction system is constructing for now. In this situation, we have to validate that EDMP was produced by these two enzymes (TDH and KBL) with other data. Based on the following results, we believe that our proposed mechanism is reasonable.

1a. By following the comment No.4 of reviewer #1, we can prove that the products of CnTDH and CnKBL are incorporated into EDMP from a shift of their *m/z* value (Fig. 2). All carbon and nitrogen atoms of EDMP derived from isotope-labeled L-Thr.

1b. Strain of *Cupriavidus necator* can generate EDMP by supplying L-Thr under oligotrophic condition (Fig. 3).

1c. Incorporation of aminoacetone produced by TDH into alkylpyrazines was reported by other researchers (Zhang et al., *AEM*, 2019, **85**, e01807-19, and Papenfort et al., *Nat. Chem. Biol.*, 2017, **13**, 551-557). Although they reported that alkylpyrazines were produced from metabolites of L-Thr and

glucose or amino acids (Zhang et al., *AEM*, 2019, **85**, e01807-19, and Papenfort et al., *Nat. Chem. Biol.*, 2017, **13**, 551-557), they did not report the stoichiometry because of above mentioned technical hurdles.

Simultaneously, we admit that the model represented in Fig. 6 in previous version cannot be validated only from currently available data. So to reflect this point, we modified the manuscript as follows.

(After) a. We revised the Figures 7a and Fig. 8 (6 in previous version) and added Figures 2 and 3.

b. The rough conversion rate of EDMP in vitro experiment: “As the concentration of CoA decreases, the unknown compound of 137.1 m/z increased (Fig. 1d, red, blue and green lines). Curiously, the production of Gly was still confirmed without CoA (Fig. 1c, green line). Besides, these reactants had a strong nutty aroma. Judging from the m/z of LC-MS and mass spectrum of GC-MS analyses (Fig. 1d and Supplementary Fig. 2), this compound was identified as EDMP. Estimates from the TIC showed that about 20-25% of the L-Thr was converted to EDMP at no or 100 μM CoA concentrations..” (lines 309-315, p. 15)

c. Following sentences were added to show that the project to apply our proposed reaction system to synthesize alkyipyrazines: “The next challenge is to optimize the synthetic method to achieve maximum yields of alkyipyrazines utilizing L-Thr as substrates. We are now trying to construct the method by combining chemo-enzymatic reactions.” (Conclusion section, lines 512-515 p. 24)

2. *Is the TA activity high enough to consider its primary role as TA rather than KBL under physiological conditions? Kinetic analysis of KBL may help understanding this.*

Response: Accordingly, we determined enzyme kinetics parameters for KBL activity. To describe this point, we added the following modifications in the manuscript.

(After) a. Kinetics parameters of KBL activity for CnKBL were represented in Supplementary Table 4.

b. we added the following sentences to compare TA and KBL activity: “

The turnover rate for the TA activity of CnKBL was comparable to that of *Streptomyces coelicolor* TA³⁷, but was much lower than that of *E. coli* TA (EcTA)³⁸. The catalytic efficiency (k_{cat}/K_m) of CnKBL toward L-Thr was 40-fold lower than that of EcTA. Next, the KBL activity was verified. Due to the instability of AKB substrate and the difficulty in detecting products, Mukherjee *et al.* uses glycine as a substrate and detect the formation of the thiol group using Ellman’s reagent (Supplementary Fig. 7)³⁶. Therefore, the kinetic parameters of condensation activity in CnKBL were determined and compared with those of EcKBL³⁶. The k_{cat} , K_m and catalytic efficiency values of CnKBL for Gly were quite comparable to those of EcKBL. The apparent KBL activity of CnKBL is very superior to the TA activity of CnKBL. However, the K_m value of CnKBL was 4.1 times higher than that of EcKBL, suggesting that its activity is susceptible to the concentration of CoA (Supplementary Fig. 8 and Talbe4). In fact, in low nutrient conditions, the concentrations of CoA and acetyl-CoA in animal and bacterial cells have been reported to be reduced to 1.3-20 μmol^{43, 44}. These results are consistent with the EDMP production *in vivo* and discussions mentioned above. (lines 383-398, pp.18-19). Experimental procedure was also added.

3. *Physiological role of the CnTDH-CnKBL operon is poorly supported by the data. Production of EDMP by Cupriavidus bacterium demonstrates conclusive evidence.*

Response: Accordingly, we try to represent that living *C. necator* cell can generate EDMP only from L-Thr. Production of EDMP by *C. necator* was analyzed by GC-MS under eutrophic (NBRC medium, Fig. 3, black line) and oligotrophic condition (M9 medium, Fig. 3, red line), respectively. To describe the results, we added the following sentences in the manuscript: “Besides, utilizing L-Thr as a substrate, EDMP could be synthesized under physiological conditions in the strain of *C. necator*. GC-MS analysis indicated that there is a peak corresponding with EDMP after 3 days of cultivation under M9 medium containing L-Thr but not in NBRC medium (Fig. 3). This suggested that EDMP would be synthesized under oligotrophic conditions in the strain.” (lines 338-343, p. 16) Experimental procedure, Fig. 3 and some related sentences have been corrected. OK

4. *I recommend CnTDH-CnKBL reactions containing isotope-labeled threonine to clarify the 3-ethyl residue of EDMP originating from acetaldehyde.*

Response: Accordingly, we performed the CnTDH-CnKBL reactions utilizing isotope-labeled threonine, and confirmed that products of CnTDH and CnKBL are incorporated into EDMP. To describe this point, we added the following sentences in the manuscript.

(After): Next, we used [U-¹³C,¹⁵N]-L-Thr to check whether the supplied L-Thr is incorporated into EDMP by CnTDH and CnKBL. Increased units of mass (e.g.10 unit, m/z 145.1-135.1) indicates that all carbon and nitrogen atoms of EDMP derived from isotope-labeled L-Thr. (lines 323-326, p. 15) Experimental procedure, Fig. 2 and some related sentences have been corrected.

Minor comments:

1. *Line 110, Origin of the plasmid, and the bacterium should be provided.*

Response: Accordingly, we added information about origin of the plasmid, and the bacterium.

(After): Dried cells of *C. necator* (NBRC 102504) was purchased from NBRC (Biological Resource Center, NITE). Gene for *cnkbl* on the genome of *C. necator* was amplified by PCR using primers (Supplementary Table 1). The amplified gene was subcloned into pET28a vector cleaved by *NcoI/XhoI*. The prepared pET28a-cnkbl plasmids were utilized in the following experiment. (lines 110-114, p. 6)

2. *Line 310, Missing reference.*

Response: Accordingly, we added two references in the manuscript.

(After): ... is a product of TDH catalysis^{21,36}. (line 352, p. 17)

3. *Line 356, changes*

Response: Accordingly, we modified the tense of “changes” as follows.

(After): changed (line 405, p. 19)

4. *Please note molar relationship of the chemicals.*

Response: Accordingly, we added the information on molar relationships of the chemical and added the following sentences in the manuscript.

(After): Here, we predicted that the condensation of two aminoacetones and acetaldehyde would progress by following the reaction scheme as shown in Fig. 7a. Aminoacetone supplied by CnTDH would be condensed and dehydrated to 2,5-dimethyl-3,6-dihydropyrazine (DHP); the production of DHP has been already reported by other groups^{5, 16}. Through tautomerization, additional reaction of DHP and acetaldehyde, and subsequent dehydration reaction, EDMP would be produced. (lines 472-478, p. 22).

Response to Reviewer #2

We appreciate you to read our manuscript and give kind comments. Modification points were highlighted by red color in the manuscript. The responses to the comments shown as followings. *The comments/questions are shown in italicized blue text. All corrections in the manuscript are shown in red.*

Major points

As a major point concerning the mechanism for EDMP formation, author must first clarify the stoichiometry of the chemoenzymatic process, in which 3 equivalents of L-Thr (C4) may be converted to one eq. of EDMP (C8), one eq. of glycine (C2), and two eq. of CO₂. The designation presented in the Figure 1 is not clear as for the stoichiometry issue.

Reviewer 1 also pointed out for the concern about determination of stoichiometry of our proposed reaction. For this point, we wrote the response as shown in answer to comment 1 of reviewer 1 (Please see the answer). In summary, because of several of technical hurdles, it is hard to determine the stoichiometry accurately. However, substantial EDMP produced in vitro and in vitro under physiological concentration of CoA. Thus, we modified the manuscript with referring to the already published paper which reported production of alkylpyrazines utilizing metabolites of L-Thr and glucose or other amino acids but not to mention the stoichiometry (Zhang et al., *AEM*, 2019, **85**, e01807-19, and Papenfort et al., *Nat. Chem. Biol.*, 2017, **13**, 551-557).

It should be pointed out in my opinion that decarboxylation can be a significant part of the TDH catalysis because AKB has the structure of beta-keto-acid form, which spontaneously decomposes by releasing CO₂ off the molecule. Decarboxylation may occur before the dehydrogenated product is released from TDH because the enzyme has an Asp residue that is associated with the pyridine-N in the PLP prosthetic group. This is typically seen in aminotransferase and decarboxylation enzymes. Accordingly, the authors must clarify the discrete role of the two enzymes contributing to produce 1 eq. EDMP from 3 eq. of L-Thr substrates.

For this comment, we firstly apologize for insufficient description about TDH. TDH is NAD⁺ dependent enzyme (and not PLP-dependent enzyme) catalyzing dehydrogenation of side chain hydroxyl group of L-Thr to AKB. TDH catalyzes only dehydrogenation reaction. The product, AKB, is rapidly decarboxylated to aminoacetone and CO₂ as already reported in previous studies (Mukherjee et al., *J. Biol. Chem.*, 1987, **262**, 30, 14441-7, and Schmidt et al., *Biochemistry*, 2001, **40**, 17, 5151-

60). Transfer mechanism of AKB from TDH to KBL is essential process to produce the Acetyl-CoA, but the mechanism remains unsolved questions for now.

Summarizing previous researches and our suggested results for TDHs and KBLs, we predicted the roles of these enzymes as follows: TDH converts L-Thr to AKB, and AKB is metabolized to Gly by KBL in usual condition. However, when they are poorly nourished and the concentration of CoA is low, unstable AKB is decarboxylated to aminoacetone, and KBL converts L-Thr to acetaldehyde and Glycine. To complement this point and clarify the roles of the enzymes, we added following sentences in the manuscript.

(After) TDH converts L-Thr to AKB, and AKB is metabolized to Gly by acetyltransferase activity of KBL in usual conditions (Fig. 8). However, in poor nutrition and low CoA concentration, unstable AKB is gradually decarboxylated to aminoacetone, and lyase activity of KBL decomposes L-Thr to acetaldehyde and Glycine (Fig. 7a and 8). (Conclusion section, lines 508-512, p. 24)

Chemical reaction mechanism, besides the enzymatic catalysis, must also be discussed in detail. How two moles of aminoacetone can form pyrazines and what is the role of aldehydes in drawing the equilibrium in favorer of forward direction and to make the 3-alkyl-2,5-dimethylpyrazines.

Response: Accordingly, we revised the sentences and Fig 7a.

(After): Here, we predicted that the condensation of two aminoacetones and acetaldehyde would progress by following the reaction scheme as shown in Fig. 7a. Aminoacetone supplied by CnTDH would be condensed and dehydrated to 2,5-dimethyl-3,6-dihydropyrazine (DHP); the production of DHP has been already reported by other groups^{5, 16}. Through tautomerization, additional reaction of DHP and acetaldehyde, and subsequent dehydration reaction, EDMP would be produced. (lines 472-478, p. 22). Fig. 7a and some related sentences have been corrected.

Minor points

1. P2. Line 29, *We surmise that EDMP*

The choice of word 'surmise' is not convincing enough for usage in paper. Readers want conclusion based on lines of evidence rather than speculation.

Response: The abstract has been fully revised.

2. P2. Line 31, *Aminoacetone was supplied by TDH which converts L-Thr to aminoacetone via unstable 2-amino-3-ketobutyrate.*

This sentence is written in past passage, but the result does not represent the evidence of aminoacetone formation nor the detection of 2-amino-3-ketobutyrate. The authors must show the evidence if this sentence is experimental results.

Response: Some groups reported aminoacetone was supplied by TDH (Mukherjee et al., *J. Biol. Chem.*, 1987, **262**, 30, 14441-7, and Schmidt et al., *Biochemistry*, 2001, **40**, 17, 5151-60). So we should add these references in the manuscript to describe this point. However, we cannot add the references at page 2, because the sentence is placed in Abstract section. As an alternative, references were added to page 17.

3. P4, Line 84, *The group stated that*

The author must select an appropriate word when quoting other paper. The word 'stated' does not convey information how scientifically sufficient evidence did the previous paper contained. Was it just a speculation or was it firmly established by lines of evidence?

Response: Accordingly, we modified the word of “stated” as following.

(After): reported (line 83, p. 4)

4. P5, Line 94, *In this study, we attempted find another mechanism by which to synthesize pyrazines from L-Thr*

If the purpose of the present study was on finding a novel reaction mechanism, it requires the effort of kinetic studies for elucidating kinetic constants at every stage of reaction, but this paper even did not succeed to clarify discrete role of the two enzymes and chemical reaction scheme after the product released from enzymes. To work on mechanistic issues, work on kinetics of each enzyme reaction and chemical reaction.

Response: In this study, we try to find synthetic pathway of alkylpyrazine in bacteria utilizing only L-Thr as substrates. As pointed out by this comment, we agree that the sentence in previous manuscript cannot reflect the purpose of this study correctly. To amend this point, we modified the manuscript as followings.

(After) a. we added the sentence: we attempted to find synthetic pathway in bacteria which can produce alkylpyrazine only from L-Thr (lines 93-94, p. 5).

b. Enzyme kinetics parameters for KBL activity of CnKBL was determined (Supplementary Table 4).

c. We added the experimental data supporting that strain of *C. necator* can synthesize EDMP only from L-Thr under oligotrophic condition (p.16 and Fig. 3)

5. P5, Line 104, *bifunctional*

The reviewer can hardly agree to this phenomena as 'bifunctionality of enzyme'. This can merely be an artificial reaction that occurred in the absence of required cofactor CoASH. The author must characterize the affinity of the enzyme to CoASH to discuss whether this could happen under physiological conditions or not

Response: As pointed out, we agree that specificity constant of TA activity is lower than that of KBL activity in CnKBL. However, enzymatic properties of KBL suggest that acetyltransferase activity gradually shifts to aldolase activity in physiological condition . Simultaneously, we can obtain experimental data *in vivo* suggesting that *C. necator* and related microorganisms can synthesize EDMP under physiological conditions. To reflect these points, we amended the manuscript as follows.

(After) a. “We predicted that CnKBL catalyzes not only acetyltransferase reaction (EC 2.3.1.29) which is broadly recognized by previous studies but also another reaction. Based on other PLP-dependent enzymes, CnKBL may catalyze carbon-carbon bond cleavage reaction (EC 4.1.X.X) as a side reaction.” (lines 328-331, p.16)

b. We added the sentences to support that EDMP can be synthesized under physiological condition as described above: “Besides, utilizing L-Thr as a substrate, EDMP could be synthesized under

physiological conditions in the strain of *C. necator*. GC-MS analysis indicated that there is a peak corresponding with EDMP after 3 days of cultivation under M9 medium containing L-Thr but not in NBRC medium (Fig. 3). This suggested that EDMP would be synthesized under oligotrophic conditions in the strain.” (line 338-343, p.16)

c. We added enzyme kinetics parameters of KBL activity in CnKBL (lines 383-398, pp.18-19 and Supplementary Table 4).

6. P5, Line 110

pET28a-CnKBL, which cloned Cupriavidus

Is the plasmid subject and the verb is to clone? It is a researcher who clones a gene.

Response: Accordingly, we modified the sentence.

(After): Dried cells of *C. necator* (NBRC 102504) was purchased from NBRC (Biological Resource Center, NITE). Gene for *cnkbl* on the genome of *C. necator* was amplified by PCR using primers (Supplementary Table 1). The amplified gene was subcloned into pET28a vector cleaved by *NcoI/XhoI*. The prepared pET28a-*cnkbl* plasmids were utilized in the following experiment. (lines 110-114, p.6)

7. P6, Line 124

The culture was cultivated

The sentence is Redundant.

Response: Accordingly, we modified the sentence as followings.

(After) pET28a-*cnkbl* in BL21(DE3) was cultivated overnight at 37 °C in 5 mL of Luria-Bertani (LB) medium containing 30 µg/mL kanamycin. (lines 123-124, p.6)

8. P6, Line 129

... was applied to a Ni²⁺-Sephrose column.

What is the volume of the column? Ambiguous description.

Response: Accordingly, we added the volume of the column as following.

(After): ... a 5 mL of Ni²⁺-Sephrose column. (line 133, p.7)

9. P7, Line 135

Purity was confirmed by sodium dodecyl sulfate-polyacrylamide gel electrophoresis.

It must describe how proteins were stained on the gel after the electrophoresis.

Ambiguous description.

Response: Accordingly, we added the method of staining the gel as following.

(After): The gel was stained with coomassie brilliant blue R-250 (Wako). (lines 140-141, p.7)

10. P11, Line 225

0.05 mM CnTDH

The concentration of the CnTDH must be rather high if this is true.

The reaction was performed under the condition containing total 0.5 mL of assay buffer. So, we believe the concentration of the CnTDH is not so high. To complement this point, we added the

following modification.

(After) The chemoenzymatic reaction was performed using 0.5 mL of assay buffer F... (line 273, p.13)

11. P13, Line 281

this compound could be EDMP

The author seems to be much less confident of EDMP that was produced in the enzyme reaction. In fact, this paper has very limited evidence that supports the EDMP being produced.

In this study, we assigned the production of EDMP by m/z value of LC-MS analysis and library search of fragment ions of the compounds obtained by GC-MS analysis. To complement this point, we amended the sentence as following.

(After) Besides, these reactants had a strong nutty aroma. Judging from the m/z of LC-MS and mass spectrum of GC-MS analyses (Fig. 1d and Supplementary Fig. 2), this compound was identified as EDMP. (lines 311-314, p.15)

12. P14, Line 293

An unexpected carbon-carbon bond cleavage reaction

There is not an unexpected C-C bond cleavage in the enzyme reaction. There is either one of the two C-C bond cleavage clearly expected according to PLP-dependent enzymes.

As pointed out by this comment, we admit that the description is inappropriate. To amend this point, we modified the sentence as following.

(After) We predicted that CnKBL catalyzes not only acetyltransferase reaction (EC 2.3.1.29) which is broadly recognized by previous studies but also another reaction. Based on other PLP-dependent enzymes, CnKBL may catalyze carbon-carbon bond cleavage reaction (EC 4.1.X.X) as a side reaction. (line 328-331, p.16)

13. P15, Line 319

And therefore we hypothesized that CnKBL has TA activity.

The TA activity of CnKBL depends on the absence or low concentration of CoASH according to the results in Figure 1 b-d. It is recommended to measure the K_m value for CoASH for the subject CnKBL, and discuss whether catalysis could promote aldolase reaction or acetyltransferase reaction under physiological condition.

Response: Accordingly, we measured the K_m value for Acetyl-CoA in lieu of CoA. This is because we cannot utilize fragile substrates, AKB. To answer this comment, we added the following descriptions in the manuscript.

(After) The catalytic efficiency (k_{cat}/K_m) of CnKBL toward L-Thr was 40-fold lower than that of EcTA. Next, the KBL activity was verified. Due to the instability of AKB substrate and the difficulty in detecting products, Mukherjee *et al.* uses glycine as a substrate and detect the formation of the thiol group using Ellman's reagent (Supplementary Fig. 7)³⁶. Therefore, the kinetic parameters of condensation activity in CnKBL were determined and compared with those of EcKBL³⁶ The k_{cat} , K_m

and catalytic efficiency values of CnKBL for Gly were quite comparable to those of EcKBL. The apparent KBL activity of CnKBL is very superior to the TA activity of CnKBL. However, the K_m value of CnKBL was 4.1 times higher than that of EcKBL, suggesting that its activity is susceptible to the concentration of CoA (Supplementary Fig. 8 and Table 4). In fact, in low nutrient conditions, the concentrations of CoA and acetyl-CoA in animal and bacterial cells have been reported to be reduced to 1.3-20 μmol ^{43, 44}. These results are consistent with the EDMP production *in vivo* and discussions mentioned above. (lines 385-398, pp. 18-19)

14. P17-18

Crystallographic studies certainly suggested putative catalytic role of residues for aldolase reaction on PLP-bound Thr. How about the acetyltransferase mechanism? Was there an open cavity for CoA to be bound for accepting acetyl group near the active site?

Response: Accordingly, we added the following sentences in the manuscript.

(After): Here, the reaction mechanism of KBL activity for CnKBL is identical to that of the already reported EcKBL^{21, 39}. As supporting data, the sequence identity between EcKBL and CnKBL is high (61%), and active site residues, S187 (S185 in EcKBL) and H215 (H213 in EcKBL), are conserved to each other (Supplementary Fig. 9). Furthermore, CnKBL has a cavity to bind CoA to the active site as well as EcKBL²¹. (lines 407-412, pp. 19)

15. P19, Line 416

Give the yield of the variants produced in combination of diverse aldehydes.

As pointed out by this comment, we agree that it is better to estimate product yield of each alkylpyrazines. However, we cannot optimize reaction methods to achieve in maximum yield of alkylpyrazines. The product yield should be determined after completion of the optimization because we may underestimate the product yield in the current situation. To mention this point, we added the following description in the manuscript.

(After) The next challenge is to optimize the synthetic method to achieve maximum yields of alkylpyrazines utilizing L-Thr as substrates. We are now trying to construct the method by combining chemo-enzymatic reactions. (conclusion section, line 512-516, p.24)

Reviewer's comments:

Reviewer #1 (Remarks to the Author):

Unfortunately, this version still lacks stoichiometry of the reaction, and does not meet the basic requirement of chemistry. I do not recommend this paper for the publication in this journal.

Reviewer #2 (Remarks to the Author):

Thank you for considering comments and suggestions for revising the manuscript. It is unfortunate that stoichiometric analysis was hard to perform, but this may be spared for your continuing research.

Even though I am satisfied with the scientific contents in the revised manuscript, I still feel stress while reading the text. I presume it is for the reason of Japan-english style writing. Sentences and compositions are awkward, and I wish the manuscript be submitted to some English-editing service, hopefully, that arrange two editing specialists, one for science and the other for style-writing. Then your paper will be smooth for every native readers, and highly evaluated.

sincerely

Reviewer #3 (Remarks to the Author):

I read through the manuscript and then looked at the review comments and rebuttal. The authors do claim in their abstract that substantial amounts of EDMP can be synthesized, yet can't quantify it, and I'm not sure other than molecular mass they have evidence it exists e.g. no standard used in LCMS/GCMS, no NMR data. The papers they refer to as also having technical limitations preventing stoichiometric analysis also have a very different focus. Considering that this study relates to a novel enzyme with novel activities and mechanisms, and the authors admit that they can't prove their mechanism with the data they have, I don't think that the study is complete enough for publication in this journal.

In addition to responding to the reviewers' comments, we corrected some sentences that were difficult to read. Please note that the updated manuscript includes the following corrections.

- NMR analysis of chemoenzymatically produced EDMP was performed. Supplementary Figure 3 has been added.

- The yield of EDMP produced by CnTDH and CnKBL from L-Thr was determined. Supplementary Figure 4 have been added.

- Comparison of EDMP and DMP productions under non-enzymatic and enzymatic conditions was added in the result & discussion section. Figure 13, Figure 14 and Supplementary Table 6 have been added.

As most people know, pyrazine is an aroma component produced by heating, as is typical in coffee and peanuts. However, in the chemoenzymatic reaction that we found in this study, the rate of EDMP

formation clearly decreases when the reaction condition exceeds a medium temperature. Supplementary Table 6, added in this revision, clearly illustrate this point. This may be evidence that some pyrazines can be synthesized under biological conditions and can express physiological functions.

Response to reviewer #1

We appreciate you to read our manuscript and give kind comments. Modification points were highlighted by red color in the manuscript. The responses to the comments shown as followings. *The comments/questions are shown in italicized blue text. All corrections in the manuscript are shown in red.*

Comments:

1. *Unfortunately, this version still lacks stoichiometry of the reaction, and does not meet the basic requirement of chemistry.*

Response: As shown in Supplementary Table 6 and new section named “Yields of EDMP and DMP under chemical or chemoenzymatic reactions.” The condensation reaction between aminoacetone and acetaldehyde generate EDMP and DMP. This suggested that we cannot indicate accurate stoichiometry with experiment because the DMP would be formed. In this situation, assignment of reaction intermediate would be an alternative to support the proposed reaction mechanism, here, DHP is some instable compound and therefore, we could not assign DHP directly. Real-time NMR experiments were also tried, but as in previous reports, the intermediates could not be detected. However, table 6 clearly shows that the intermediates is slowly converted to DMP in a maximum yield of about 10%. In the presence of aldehyde, EDMP is produced via nucleophilic addition reaction, with a maximum yield of 16.2% under physiological temperature. The fact that the yield exceeded 20% in the chemoenzymatic reaction may be explained by the similarity of the optimal reaction temperature between the enzymatic and chemical reactions and the accelerating effect of TDH and KBL on EDMP synthesis. To illustrate these points, we added the following description to the manuscript.

(After): ... and NMR analysis (Supplementary Fig. 3), this compound was identified as EDMP. HPLC analysis showed that the yield of EDMP from L-Thr was approximately 4%. The yield of EDMP was calculated from Supplementary Fig. 4. (lines 320–323, P15–16)

Comparison of EDMP production under chemical and chemoenzymatic reactions

Structural and functional analysis of CnTDH and CnKBL suggested that EDMP could be synthesized only from L-Thr with chemoenzymatic reaction. Aminoacetone and acetaldehyde, precursors to generate EDMP, are supplied by enzymatic reactions, and chemical reactions progress their condensation. The next challenge is to optimize the reaction conditions to maximize the amount of EDMP produced. It is expected that reaction temperature and timing of addition of precursor would affect the rate of EDMP production.

Firstly, we estimated the optimal temperature for the maximum synthesis of EDMP by changing the reaction temperature (Entry 1–4 in Supplementary Table 6). The yield of EDMP was as high as 16.2% under the chemical reaction condition of 30 °C. Notably, this yield gradually decreased as the

temperature increased. In the Maillard reaction, pyrazine is generated at high temperatures^{9, 10}. The fact that the chemical condensation reaction of aminoacetone and acetaldehyde proceeds under such moderate conditions may have implications for the identification of pyrazine compounds *in vivo* (Fig. 3).

The synthesis mechanism of EDMP is shown in Fig. 7a. First, as is well known, aminoacetone is supplied from L-Thr by TDH. Next, the two aminoacetones condense to 2,5-dimethyl-3,6-dihydropyrazine (DHP). Finally, DHP tautomerizes and undergoes nucleophilic addition to acetaldehyde to synthesize EDMP. Other research groups reported that the DHP is easily oxidized under mild condition to form 2,5-dimethylpyrazine (DMP)^{42, 43}, suggesting that immediate reaction of DHP with acetaldehyde increases the production of EDMP. To confirm this point, we compared production rates of EDMP and DMP by changing the timing of acetaldehyde addition (Supplementary Table 6). As expected, yield of EDMP is maximized at the condition which mixed acetaldehyde and aminoacetone simultaneously (16.2%, entry 5 in Supplementary Table 6). By delaying the timing of addition, the production rate gradually decreased. When acetaldehyde was added after 12 hours of pre-reaction, the EDMP production rate was minimal (Entry 9 in Supplementary Table 6). On the other hand, the opposite tendency was observed for the production rate of DMP (Entry 5–9 in Supplementary Table 6 and Supplementary Fig. 13 and 14).

Considering the time-dependent changes in the production rates of EDMP and DMP, it was found that the reaction intermediate, DHP, was stable for a certain time and that the reaction of DHP and acetaldehyde at physiological temperature was important for the chemoenzymatic synthesis of EDMP. When the precursor was supplied from L-Thr by the enzymes, the yield of EDMP was increased up to 20.2%.

(lines 474–511, P22–24)

Experimental procedures related to Supplementary Fig. 3 and 4, and some related sentences have been corrected.

Response to reviewer #2

We deeply appreciate the reviewer both for his/ her patience, valuable time and precise comments/questions. Our answers were written in followings. Modification points were highlighted by red color in the manuscript. *The comments/questions are shown in italicized blue text. All corrections in the manuscript are shown in red.*

Comments:

1. *Even though I am satisfied with the scientific contents in the revised manuscript, I still feel stress while reading the text. I presume it is for the reason of Japan-english style writing. Sentences and compositions are awkward, and I wish the manuscript be submitted to some English-editing service, hopefully, that arrange two editing specialists, one for science and the other for style-writing. Then your paper will be smooth for every native readers, and highly evaluated.*

Response: Accordingly, we submitted the manuscript to English-editing service.

Response to reviewer #3

Thank you for giving some critical comments with carefully reading of our manuscript. Our answers were written in followings. Modification points were highlighted by red color in the manuscript. *The comments/questions are shown in italicized blue text. All corrections in the manuscript are shown in red.*

Comments:

1. *The authors do claim in their abstract that substantial amounts of EDMP can be synthesized, yet can't quantify it.*

Response: We calculated the yield of EDMP produced by CnTDH and CnKBL from L-Thr. To describe this point, we added the following description in the manuscript.

(After): ... HPLC analysis showed that the yield of EDMP from L-Thr was approximately 4%. The yield of EDMP was calculated from Supplementary Fig. 4. (lines 321-323, P15–16)

As expected, yield of EDMP is maximized at the condition which mixed acetaldehyde and aminoacetone simultaneously (16.2%, entry 5 in Supplementary Table 6).

(lines 498-500, P24)

When the precursor was supplied from L-Thr by the enzymes, the yield of EDMP was increased up to 20.2%.

(lines 509-511, P24)

Experimental procedure related to Supplementary Fig. 4 and some related sentences have been corrected.

2. *and I'm not sure other than molecular mass they have evidence it exists e.g. no standard used in LCMS/GCMS, no NMR data.*

Response: We added the GC-MS data using EDMP standard (Supplementary Fig. 2) and NMR data

of EDMP produced by CnTDH and CnKBL from L-Thr (Supplementary Fig. 3). To describe this point, we added the following description in the manuscript:

(After): Judging from the m/z of LC-MS (Fig. 1d), mass spectrum of GC-MS (Supplementary Fig. 2), and NMR analysis (Supplementary Fig. 3), this compound was identified as EDMP. (lines 319-321, P15)

Experimental procedures related to Supplementary Fig. 2 and 3, and some related sentences have been corrected.

3. The papers they refer to as also having technical limitations preventing stoichiometric analysis also have a very different focus.

Response: Reviewer 1 also pointed out for the concern about determination of stoichiometry of our proposed reaction. For this point, we wrote the response as shown in answer to comment 1 of reviewer 1 (Please see the answer). In summary, we estimated the yield of EDMP.

Reviewer's comments:

Reviewer #1 (Remarks to the Author):

The authors added new experiments for conversion yields of EDMP (Supplementary table 6), which lost impact of the paper.

They shows only 4% and minor contribution of TDH/KBL, and no explanation in major reaction product originated from around 80% of Thr. I feel very small advance of the use of TDH/KBL for applied/process chemistry for pyrazine production.

Response to reviewer #1

We appreciate you to read our manuscript and give kind comments. Modification points were highlighted by red color in the manuscript. The responses to the comments shown as followings. *The comments/questions are shown in italicized blue text.*

Comments:

- 1. The authors added new experiments for conversion yields of EDMP (Supplementary table 6), which lost impact of the paper. They shows only 4% and minor contribution of TDH/KBL, and no explanation in major reaction product originated from around 80% of Thr. I feel very small advance of the use of TDH/KBL for applied/process chemistry for pyrazine production.*

Response: We found that TDH/KBL produces 20.2% EDMP from L-Thr and DMP as a byproduct. As you pointed out, a large part of it is converted and degraded to other by-products. Certainly, in terms of pyrazine material production, the advantages of this study may be small. However, the fact that 20% of the coffee-like aroma can be produced from amino acids in water is very impressive. We would like to improve the yield in the future. For example, if the reaction is carried out in aqueous two-phase system, the yield may be improved.